# The Cost of Information: Phase Transitions in Contextual Bandits with Paid Observations

Xueping Gong [1]   Jiheng Zhang [2]

## Abstract

We study contextual bandits with paid observations, where the learner actively chooses which actions to observe at a given cost in each round, with the goal of minimizing total regret that jointly accounts for learning loss and observation expenditure. We develop a near-optimal algorithm for adversarial environments and show that even small observation costs fundamentally raise the minimax regret order. We further uncover a novel phase transition under a free observation budget: below a critical threshold, free observations only reduce total cost without improving the regret rate; above it, asymptotic improvements become possible. To exploit this phenomenon, we design a meta-controller that adaptively switches between strategies to achieve near-optimal performance across all budget regimes. To handle large or infinite policy spaces, we also propose an oracle-efficient algorithm under a function approximation framework that maintains rigorous guarantees with computational efficiency. Our analysis also connects to related problems including switching costs, budgeted constraints, model misspecification, and knapsack bandits. Numerical experiments validate our theoretical findings.

## 1. Introduction

The contextual bandits problem (Chu et al., 2011; Abbasi-Yadkori et al., 2011; Foster et al., 2018; Agarwal et al., 2012; Foster & Rakhlin, 2020a; Gong et al., 2023) is a well-established framework for studying sequential decision-making with rich contextual information. It has gained significant attention due to its wide range of applications, including scheduling, dynamic pricing, packet routing, online auctions, e-commerce, and matching markets (Lattimore & Szepesvári, 2020). Classical contextual bandit algorithms have been highly successful in tasks where feedback is readily available and easy to obtain.

However, there are scenarios where the learner must pay for feedback, which can significantly impact the learning process. In such scenarios, the learner may acquire additional feedback at a prescribed cost, introducing a trade-off between information gain and expenditure. For instance, in IoT networks, high-fidelity sensor data collection incurs energy costs; in conversational AI, hiring human evaluators to assess response quality entails financial expenses; and in recommendation systems or medical insurance, monitoring long-term user satisfaction or service quality involves operational overhead. Similar challenges manifest in finance through the procurement of proprietary market data, in robotics via sensor calibration, and in personalized education through detailed student performance assessments. Despite the critical influence of such observation costs on strategic decision-making, they remain a largely understudied aspect of the contextual bandits literature. To bridge this gap, we explore a realistic setting where obtaining accurate feedback information involves costly interactions.

Within this framework, the learner faces a fundamental trade-off over the $T$ rounds of interaction: whether to pay a cost to observe the loss of certain actions. Specifically, in each round $t \in [T]$, after observing the context $x_t$ and the costs $\{c_t(a)\}_{a \in \mathcal{A}}$ for each action in $\mathcal{A}$, the learner selects a policy from the policy class $\Pi$ to choose an action $a_t$, and additionally selects a subset of actions $O_t \subset \mathcal{A}$ to observe. For each action $a \in O_t$, the learner pays a cost of $c_t(a)$ and receives the corresponding loss value $\ell_t(a)$; losses of actions outside $O_t$ remain unobserved.

This setting significantly extends the classical exploration-exploitation trade-off in contextual bandits: the learner must balance two key objectives—reducing uncertainty via observation, and minimizing the cumulative cost incurred by such observations. Given that both contexts and losses may be adversarial, the learner faces a dual challenge: strategically managing observation costs while tracking the potentially

[1]Department of Management, Xiamen University, Xiamen, China [2]Department of Industrial Engineering & Decision Analytics, Hong Kong University of Science and Technology, Hong Kong, China. Correspondence to: Jiheng Zhang <jiheng@ust.hk>, Xueping Gong <xgongah@xmu.edu.cn>.

*Proceedings of the 43rd International Conference on Machine Learning*, Seoul, South Korea. PMLR 306, 2026. Copyright 2026 by the author(s).

evolving optimal action. Expanding the observation set $O_t$ offers clear benefits: it enables more accurate estimates of the loss function and can enhance long-term decision quality. However, this advantage comes with a direct trade-off: the associated cost $\sum_{a \in O_t} c_t(a)$ contributes directly to the overall regret. Therefore, at each round, the learner must dynamically assess whether the potential information gain from observing a particular action $a$ justifies the immediate cost $c_t(a)$. This cost-aware observation strategy introduces a new layer of resource-allocation complexity to online learning, where efficiently tracking the context-dependent optimal action becomes paramount.

**Contributions.** Our paper focuses on the *contextual bandits with paid observations* problem, addressing the limitation of traditional models that ignore observation costs.

1. **Algorithm Development:** We extend prior work on multi-armed bandits with paid observations (Seldin et al., 2014) to the contextual setting, proposing Algorithm 1. Unlike prior methods lacking rigorous guarantees or restricted to narrow policy spaces (Tucker et al., 2022), this algorithm handles general policy spaces with formal theoretical analysis, as shown in Theorem 2.1. We establish its minimax optimality by proving a tight regret lower bound (Theorem 2.2) —showing even small observation costs drive the regret lower bound from $\Theta(T^{\frac{1}{2}})$ (cost-free optimal rate) to $\Theta(T^{\frac{2}{3}})$ (paid observation optimal rate). This sharp order shift highlights how paid observations fundamentally reshape learning complexity.

   To enhance computational efficiency, we develop an oracle-efficient algorithm, Algorithm 3, within a function approximation framework. Under a realizability assumption, we leverage an online regression oracle to enable efficiency even for infinite function spaces. Theorem 4.1 confirms that the regret upper bound aligns with the statistical complexity. Notably, while (Tucker et al., 2023) studies linear contextual bandits with paid observations, it provides no theoretical guarantees. Our work closes this gap with Algorithm 3, which supports general function spaces and is backed by rigorous theoretical analysis.

2. **Phase Transition Phenomenon:** We identify a critical phase-transition phenomenon when incorporating free observation budgets. There exists a threshold budget: above it, free observations sharply reduce the regret order; below it, their impact is negligible and fails to alter the regret order. This reveals the core trade-off between cost control and regret minimization in contextual bandits. To exploit this phenomenon, we design a meta-controller (Algorithm 2) that adaptively selects strategies based on the budget regime. As proven in

Theorem 3.2, this meta-controller matches the theoretical lower bound (Theorem 3.1) for the free budget setting, ensuring near-optimal performance across all budget scenarios.

### 1.1. Related work

**Contextual bandits.** Our work is closely related to the recent trend of designing efficient algorithms for contextual bandits. Researchers have designed algorithms tailored to specific function space structures, developing efficient contextual bandits algorithms via reduction to regression oracles (Foster et al., 2018; Agarwal et al., 2012; Foster & Rakhlin, 2020a; Gong et al., 2023; Simchi-Levi & Xu, 2021). Notably, these approaches do not account for paid observations, a gap that our algorithm aims to fill by addressing this challenge and demonstrating a significant difference in regret orders.

**Graph feedback.** Additionally, our research is relevant to the domain of online learning with side information modeled by feedback graphs. Methods such as UCB (Lykouris et al., 2020; Caron et al., 2012), TS (Lykouris et al., 2020), EXP (Rouyer et al., 2022; Chen et al., 2021; Alon et al., 2015; Cohen et al., 2016), and IDS (Liu et al., 2018), along with their variants, have been developed for the non-contextual setting. (Zhang et al., 2024) and (Chu et al., 2011) have integrated graph feedback with contextual information. While graph-structured feedback inherently benefits online learning and can lead to a decrease in regret order, as supported by Lemma 2.2, the introduction of costs for these additional observations can significantly elevate the regret order from $\tilde{\Theta}(T^{\frac{1}{2}})$ to $\tilde{\Theta}(T^{\frac{2}{3}})$. For example, (Dekel et al., 2014) consider the switching costs in the graph feedback setting, and the regret scales as $\tilde{\Theta}(\gamma(G)^{\frac{1}{3}} T^{\frac{2}{3}})$, with $\gamma(G)$ denoting the dominating number of the given graph.

**Switching cost.** Our setting shares similarities with bandit problems involving switching costs, as discussed in previous works such as (Dekel et al., 2014; Rangi & Franceschetti, 2019; Cheng et al., 2023). Interestingly, we can demonstrate a transformation of a bandit problem with switching costs into a simplified version within our setting.

In the transformed scenario, the learner incurs a cost $c$ every time it switches to a new action. We can assign the cost $c$ to all actions $a \neq a_{t-1}$; otherwise, we set the cost to 0. This conversion effectively treats the switching cost as the price paid to obtain observations of the selected action. Consequently, our algorithm can be modified to handle the adversarial MAB problem with switching costs, resulting in regrets of $\mathcal{O}((c|\mathcal{A}| \ln |\mathcal{A}|)^{\frac{1}{3}} T^{\frac{2}{3}})$. This regret order aligns with the findings presented in (Dekel et al., 2014; Rangi & Franceschetti, 2019; Altschuler & Talwar, 2018). It is important to note that the lower bound for MAB with switching costs is $\Omega((c|\mathcal{A}|)^{\frac{1}{3}} T^{\frac{2}{3}})$ in (Dekel et al., 2014).

Actually, the introduction of the freedom to remain at one action does not fundamentally alter the inherent difficulty level, as highlighted by Theorem 2.2.

**Model misspecification.** The work by (Neu & Olkhovskaya, 2020) addresses the adversarial linear bandit problem with model misspecification and presents a robust algorithm to handle this scenario. In our framework, as mentioned in the function approximation setting, our policy space can be induced by linear spaces. Interestingly, the regret caused by model misspecification can be interpreted as a form of paid cost. Each observation incurs an additional cost of $\epsilon$ due to model misspecification. Consequently, our result provides an informal explanation for the second leading term being of order $\mathcal{O}(T^{\frac{2}{3}})$ in (Neu & Olkhovskaya, 2020). This additional term accounts for the cost associated with model misspecification, which aligns with the concept of paid costs in our framework.

**Knapsack.** Bandit with knapsack refers to an online learning problem with constraints. In classical knapsack problems, constraints and regrets are typically treated independently. For instance, (Sun et al., 2017) achieve a regret of $\tilde{\mathcal{O}}(\sqrt{|\mathcal{A}|T \ln |\Pi|})$ while incurring costs of $\mathcal{O}(T^{\frac{3}{4}})$. Additionally, (Chzhen et al., 2024) aim for a total cost of $\mathcal{O}(T)$ while accepting regrets of $\tilde{\mathcal{O}}(\sqrt{|\mathcal{A}|T})$.

However, when we combine the two factors, a trade-off arises between the cost of observations and the learning regrets. The constraints can be viewed as a form of paid cost for obtaining observations. As we have demonstrated in Theorem 2.2, the overall costs will be at least $\Omega(T^{\frac{2}{3}})$. This indicates that there exists an inherent trade-off between the observation cost and the learning regrets in the bandit with knapsack setting.

## 2. Paid Observations

This section introduces a setting in which the learner can actively acquire additional observations at a cost. We refer to this as the *paid observation* setting. We show that our algorithm naturally generalizes both the standard cost-free setting and the paid observation setting, providing a unified approach that handles both regimes.

### 2.1. Adversarial Setting

We work within the adversarial contextual bandits framework. The learner has access to a finite action set $\mathcal{A}$ and a finite policy class $\Pi = \{\pi : \mathcal{X} \to \mathcal{A}\}$, which maps contexts from $\mathcal{X}$ (possibly infinite) to actions. The problem is adversarial: no stochastic assumptions (such as i.i.d. generation) are imposed on how the contexts and losses are generated.

The interaction takes place over $T$ rounds. At each round $t$, the adversary chooses a context $x_t \in \mathcal{X}$ and a set of losses

$\{\ell_t(a) \in [0, 1]\}_{a \in \mathcal{A}}$. Crucially, the adversary is assumed to be non-oblivious, as formalized below.

**Assumption 2.1** (Nonoblivious Adversary)**.** The adversary is nonoblivious, meaning that both the context $x_t$ and the loss vector $\{\ell_t(a)\}_{a \in \mathcal{A}}$ can depend on the entire history of interactions $\mathcal{H}_{t-1}$ up to round $t-1$.

Under this assumption, the loss $\ell_t(a)$ is deterministic given the history, so $\mathbb{E}[\ell_t(a) \mid \mathcal{H}_{t-1}] = \ell_t(a)$ for all $a$. Non-oblivious adversaries are strictly more powerful and lead to more challenging learning problems than oblivious adversaries (Dekel et al., 2014).

In addition to the context $x_t$, the learner also receives a cost vector $\{c_t(a)\}_{a \in \mathcal{A}}$ before making decisions at round $t$. To ensure that the regret remains bounded, we assume that the observation costs are uniformly bounded, even when chosen adversarially.

**Assumption 2.2** (Paid Observation)**.** At round $t$, the environment provides a cost vector $\{c_t(a)\}_{a \in \mathcal{A}}$. The learner may choose a subset $O_t \subseteq \mathcal{A}$ of actions to observe, by paying a cost of $\sum_{a \in O_t} c_t(a)$. In return, the learner observes $\ell_t(a)$ for all $a \in O_t$.

This framework models many real-world applications. For instance, in ad placement or content moderation, systems often decide whether to acquire costly human feedback to improve performance, creating a trade-off between labeling cost and accuracy. Additional motivating examples are provided in Section A.

At round $t$, the learner observes the context $x_t$ and cost vector $\{c_t(a)\}$, selects an action $a_t$, chooses an observation set $O_t \subseteq \mathcal{A}$ (paying $\sum_{a \in O_t} c_t(a)$), and then observes $\ell_t(a)$ for all $a \in O_t$. The filtration $\mathcal{H}_t$ in Assumption 2.1 is defined as $\sigma\big(x_s, a_s, \{c_s(a)\}_{a \in \mathcal{A}}, \{\ell_s(a)\}_{a \in O_s} : s \leq t\big)$.

The total cost incurred in round $t$ is given by: $\ell_t(a_t) + \sum_{a \in O_t} c_t(a)$. It is important to note that $\ell_t(a_t)$ might not be observed. This is because the action $a_t$ selected by the learner might not be included in the observation set $O_t$, and thus its loss cannot be retrieved.

The cumulative regret $\mathfrak{R}(T)$ measures the difference between the cumulative loss of the learner (including observation costs) and that of the best fixed policy $\pi^* \in \Pi$ in hindsight, defined as $\pi^* = \arg\min_{\pi \in \Pi} \sum_{t=1}^{T} \ell_t(\pi(x_t))$. Formally, the decomposition is given by:

$$\mathfrak{R}(T) = \mathbb{E}\left[\underbrace{\sum_{t=1}^{T} (\ell_t(a_t) - \ell_t(\pi^*(x_t)))}_{\text{Learning Performance}} + \underbrace{\sum_{t=1}^{T} \sum_{a \in O_t} c_t(a)}_{\text{Observation Cost}}\right],$$

(1)

where the expectation is taken over the randomness in the algorithm.

We aim to address the following questions: *Q1: When*

*is it beneficial to request observations? Q2: How many observations should the learner acquire?*

## 2.2. Algorithm Design

This section addresses the challenge of balancing learning performance and information cost in this setting. Building on (Seldin et al., 2014), Algorithm 1 introduces a novel adaptive observation mechanism that selectively acquires loss feedback based on both informational value and cost. This design enables loss observations for actions that are either frequently sampled (high $q_t(a)$) or inexpensive to observe (low $c_t(a)$), thereby formalizing the trade-off between information gain and cost efficiency.

---

**Algorithm 1** Adversarial Contextual Algorithm with Paid Observations

---

**Input:** Action set $\mathcal{A}$, non-increasing learning rate $\eta_t$, policy space $\Pi$, hyperparameter $\gamma$

1: **for** round $t = 1, 2, \cdots$ **do**
2:     Adversary generates a hidden loss set $\{\ell_t(a)\}_{a \in \mathcal{A}}$ and reveals a context $x_t$
3:     Observe the observation costs $c_t(a)$ for all $a \in \mathcal{A}$
4:     Compute the weight $w_t(\pi) = \exp\left(-\eta_t \sum_{s=1}^{t-1} \hat{\ell}_s(\pi(x_s))\right)$
5:     Compute the sampling probability $q_t(a) = \frac{(1-\gamma)\sum_{\pi \in \Pi : \pi(x_t)=a} w_t(\pi)}{\sum_{\pi \in \Pi} w_t(\pi)} + \frac{\gamma}{|\mathcal{A}|}$
6:     Sample an action $a_t$ according to $q_t$
7:     Compute the observation probability $p_t(a) = \min\left\{1, \sqrt{\frac{\eta_t q_t(a)}{c_t(a)}}\right\}$
8:     For each $a$, put $a$ into the observation set $O_t$ with probability $p_t(a)$
9:     Pay the total cost $\sum_{a \in O_t} c_t(a)$ to observe the losses $\ell_t(a)$ of actions in $O_t$
10:    Update $\hat{\ell}_t(a) = \mathbb{I}\{a \in O_t\} \frac{\ell_t(a)}{p_t(a)}, \forall a \in \mathcal{A}$
11: **end for**

---

Algorithm 1 constructs loss estimators via importance sampling (unbiased; see Lemma C.1) and uses them to guide action selection. Instead of applying exponential weights directly over actions, it weights policies by their cumulative historical losses, then aggregates these weights into sampling probabilities that adapt to both history and context.

## 2.3. Regret Upper Bound

Lemma 2.1 provides an analytical framework for understanding the regret upper bound of our algorithm in terms of the parameters $\eta_t$, $q_t(a)$, and $p_t(a)$. This result offers important insights into the role of the observation cost. In particular, since the expected cost of observations at round $t$ is $\sum_{a \in \mathcal{A}} p_t(a) c_t(a)$, the lemma also guides the design of the observation probabilities $p_t(a)$, thereby facilitating the

optimization of the cost-performance trade-off.

**Lemma 2.1.** If $\gamma \in (0, 1)$ and $\eta_t$ is a non-increasing sequence, then the following inequality holds $\mathbb{E}\left[\sum_{t=1}^{T} \sum_{a \in \mathcal{A}} (q_t(a) - \mathbb{I}\{\pi^*(x_t) = a\}) \ell_t(a)\right] \leq \mathbb{E}\left[2\gamma T + \frac{(1-\gamma)\ln|\Pi|}{\eta_{T+1}} + \sum_{t=1}^{T} \eta_t \sum_{a \in \mathcal{A}} \frac{q_t(a)}{p_t(a)}\right]$.

By incorporating observation costs into our analysis, we can derive an upper bound that effectively captures the impact of these costs. The order of the bound in Theorem 2.1 clearly illustrates how the costs influence the regret order.

**Theorem 2.1.** The regret of Algorithm 1 satisfies $\mathfrak{R}(T) \leq \frac{(1-\gamma)\ln|\Pi|}{\eta_{T+1}} + \sum_{t=1}^{T} \eta_t + \sum_{t=1}^{T} \sum_{a \in \mathcal{A}} \sqrt{\eta_t c_t(a) q_t(a)} + 2\gamma T$. If we choose $\eta_t = \frac{1}{\left(\frac{\sum_{s=1}^{t} \sqrt{\sum_{a \in \mathcal{A}} c_s(a)}}{\ln|\Pi|}\right)^{\frac{2}{3}} + \sqrt{\frac{t}{\ln|\Pi|}}}$, (which is always a non-increasing sequence), and $\gamma = \frac{1}{T}$, we have $\mathfrak{R}(T) \leq 4(\ln|\Pi|)^{\frac{1}{3}} \left(\sum_{t=1}^{T} \sqrt{\sum_{a \in \mathcal{A}} c_t(a)}\right)^{\frac{2}{3}} + 4\sqrt{T \ln|\Pi|} + 2$. In particular, if $c_t(a) = c$, then $\mathfrak{R}(T) \leq \mathcal{O}\left((c|\mathcal{A}|\ln|\Pi|)^{\frac{1}{3}} T^{\frac{2}{3}} + \sqrt{T \ln|\Pi|}\right)$.

An important insight from this theorem is that even a small observation cost can significantly increase the minimax regret order from $\tilde{\mathcal{O}}(T^{\frac{1}{2}})$ to $\tilde{\mathcal{O}}(T^{\frac{2}{3}})$. This finding highlights the crucial trade-off between learning regrets and observation costs, emphasizing the significance of considering this trade-off in real-world applications. As observation costs are commonly present in practical scenarios, understanding and managing this trade-off becomes essential for optimizing learning performance.

**Remark 2.1.** When the observation cost $c = 0$, our regret bound recovers the optimal rate $\mathcal{O}(\sqrt{T \ln|\Pi|})$ for standard adversarial contextual bandits. Here the learner can observe all losses freely, which reduces to the full-information setting, matching its known minimax rate.

## 2.4. Regret Lower Bound

In this section, we establish a lower bound for the paid observation setting, demonstrating that our algorithm achieves near-optimal performance. The following lemma provides an intuitive result for extra observations. Consider a scenario where the learner engages in the game for $MT$ rounds. (Agarwal et al., 2012) reveals that no algorithm can achieve a regret lower than $\Omega(\sqrt{|\mathcal{A}| MT \ln|\Pi|})$. Informally, only $\frac{1}{M}$ fraction of regrets contributes to the final result, which leads to $\Omega\left(\sqrt{|\mathcal{A}| T \ln(|\Pi|)/M}\right)$ lower bounds.

**Lemma 2.2.** For the adversarial contextual bandit game with $MT$ observed losses and a given algorithm, there exists a losses sequence $\{\ell_t(a), a \in \mathcal{A}\}_{t=1}^{T}$ and a contextual sequence $\{x_t\}_{t=1}^{T}$ such that $\mathbb{E}\left[\sum_{t=1}^{T} (\ell_t(a_t) - \ell_t(\pi^*(x_t)))\right] \geq \Omega\left(\sqrt{\frac{|\mathcal{A}| T \ln(|\Pi|)}{M \ln|\mathcal{A}|}}\right)$, where $\{a_t\}_{t=1}^{T}$ are actions generated by the given algorithm.

Based on Lemma 2.2, we can prove the following theorem. In this setting, the learner can acquire valuable information about losses only when it pays the observation costs. From Theorem 2.2, one can observe that there is a trade-off between exploration and paid observations. On one hand, to achieve better exploration and minimize regret, the learner needs to collect a larger number of observations to gain more information. On the other hand, since each observation incurs a cost, the learner should be cautious about excessive spending on observations. To strike a balance, the optimal strategy for the learner is to obtain $\Theta((c|\mathcal{A}|\ln|\Pi|)^{\frac{1}{3}}T^{\frac{2}{3}})$ observations. Deviating from this optimal number of observations would result in a higher regret, as $\Theta((c|\mathcal{A}|\ln|\Pi|)^{\frac{1}{3}}T^{\frac{2}{3}})$ represents the minimizer of the lower bound. Thus, by carefully managing the number of paid observations, the learner can achieve a favorable overall regret.

**Theorem 2.2.** For the adversarial contextual bandit game with uniform cost $c$ and a given algorithm, there exists an bandit instance and adversarial strategy such that

$$\mathfrak{R}(T) \geq \Omega\left(\max\left\{\sqrt{T\ln(|\Pi|)}, (c|\mathcal{A}|\ln|\Pi|)^{\frac{1}{3}}T^{\frac{2}{3}}\right\}\right).$$

## 3. Free Observation Budget

### 3.1. Lower Bound and Phase Transition

The gap between the regret bounds in the cost-free setting ($\Omega(T^{1/2})$) and the paid observation setting ($\Omega(T^{\frac{2}{3}})$) naturally raises the following question: "*How much can the regret be improved if the learner is permitted to make some free observations in addition to the paid ones?*"

We address this by introducing a *free observation budget* $B$ ($0 \leq B \leq |\mathcal{A}|T$) into the paid-observation setting. The learner may freely observe up to $B$ losses over the entire horizon; $B = 0$ recovers the standard paid-observation model. This hybrid framework allows us to explore whether combining free and paid observations can yield better regret guarantees, offering a more flexible and potentially more powerful approach.

**Theorem 3.1.** For the adversarial contextual bandit game with uniform cost $c$ and a given algorithm, there exists an bandit instance and adversarial strategy such that

$$\mathfrak{R}(T) \geq \begin{cases} \Omega\left((c|\mathcal{A}|\ln|\Pi|)^{\frac{1}{3}}(\ln|\mathcal{A}|)^{-\frac{1}{3}}T^{\frac{2}{3}} - cB\right), \\ \quad \text{if } B = \mathcal{O}(c^{-\frac{2}{3}}(|\mathcal{A}|\ln|\Pi|)^{\frac{1}{3}}(\ln|\mathcal{A}|)^{-\frac{1}{3}}T^{\frac{2}{3}}) \\ \Omega\left(T\sqrt{\frac{|\mathcal{A}|\ln(|\Pi|)}{B\ln|\mathcal{A}|}}\right), \\ \quad \text{if } B = \Omega(c^{-\frac{2}{3}}(|\mathcal{A}|\ln|\Pi|)^{\frac{1}{3}}(\ln|\mathcal{A}|)^{-\frac{1}{3}}T^{\frac{2}{3}}) \end{cases}$$

The phase-transition phenomenon of lower bounds highlights an important conclusion: in the presence of paid observations, the learner's ability to distinguish between different actions should be limited to approximately

$\tilde{\Theta}(c^{-\frac{2}{3}}(|\mathcal{A}|\ln|\Pi|)^{\frac{1}{3}}T^{\frac{2}{3}})$ observations, rather than the full $T$ observations in the standard contextual bandit setting (without costs). This limitation is a key factor contributing to the worst minimax regret in this setting.

Next, we consider the first case: $B = \tilde{\mathcal{O}}(c^{-\frac{2}{3}}(|\mathcal{A}|\ln|\Pi|)^{\frac{1}{3}}T^{\frac{2}{3}})$. In this scenario, one might hope to achieve a smaller regret (compared to the paid observation setting) through the utilization of additional free observations. However, we will demonstrate that, unfortunately, the benefits gained from these additional observations are negligible in terms of improving the regret order. Consequently, the previous bound of $\tilde{\Omega}((c|\mathcal{A}|\ln|\Pi|)^{\frac{1}{3}}T^{\frac{2}{3}})$ remains unchanged. The extra $B$ observations primarily serve to reduce the cost of required observations, rather than significantly impacting the regret.

In contrast to the previous case, where the budget is relatively small, the presence of a larger budget $B = \tilde{\Omega}(c^{-\frac{2}{3}}(|\mathcal{A}|\ln|\Pi|)^{\frac{1}{3}}T^{\frac{2}{3}})$ allows for more exploration through the extra observations, which do not incur any additional costs. Consequently, this leads to a smaller regret, with the budget $B$ dominating the regret in the lower bound. In other words, by leveraging the additional free observations, one can effectively reduce the regret through exploration without incurring any extra costs.

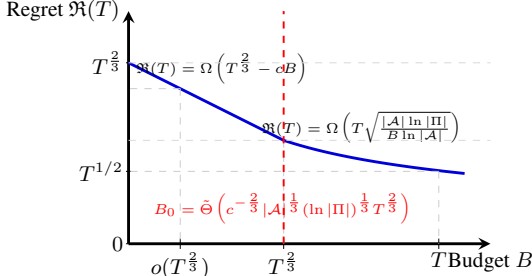

*Figure 1.* Phase transition of regret with respect to budget $B$

Figure 1 illustrates the phase transition of the cumulative regret $\mathfrak{R}(T)$ with respect to the budget $B$ for the adversarial contextual bandit game, which is derived from Theorem 3.1. The plot explicitly captures two distinct regimes of regret behavior, separated by a critical budget threshold $B_0 = \tilde{\Theta}\left(c^{-\frac{2}{3}}|\mathcal{A}|^{\frac{1}{3}}(\ln|\Pi|)^{\frac{1}{3}}T^{\frac{2}{3}}\right)$ (marked by the vertical red dashed line).

### 3.2. Algorithm Design and Upper Bound

To complete the framework, we now introduce an algorithm that achieves near-optimal regret in both budget regimes. Building upon the lower bound derived in Theorem 3.1, we design a meta-controller that dynamically selects algorithms according to the available budget $B$.

The meta-controller compares the budget $B$ against a theoretical threshold obtained from the lower bound analysis.

Depending on whether the budget is sufficient, it deploys one of two specialized variants of the adversarial algorithm, each designed for a specific budget condition. This strategy ensures the budget is used efficiently while matching the regret lower bounds.

When $B = \Omega(c^{-\frac{2}{3}}(|\mathcal{A}| \ln |\Pi|)^{\frac{1}{3}} T^{\frac{2}{3}})$, we propose Algorithm 4 in Section E. The algorithm begins by allocating the budget $B$ to generate the sequence of observable action sets $\{O_t\}_{t=1}^{T}$ through Algorithm 5 (in Appendix for conciseness). Our allocation rule ensures the observation probability $p_t(a)$ is equal to $\frac{B}{|\mathcal{A}|T}$. Then the learning rate $\eta_t$ is set proportionally to $\sqrt{B/(|\mathcal{A}| \ln |\Pi|)}$, which balances exploration and exploitation given the budget constraint.

We then consider the scenario where the budget is limited, specifically when $B = \mathcal{O}\left(c^{-\frac{2}{3}}(|\mathcal{A}| \ln |\Pi|)^{\frac{1}{3}} T^{\frac{2}{3}}\right)$. The meta-controller invokes Algorithm 6, which maintains a running budget $B_t$ that is decremented by the number of observations made in each round. When the stochastically generated observation set $O_t$ exceeds the remaining budget, the algorithm still need to pay for the excess observations.

---

**Algorithm 2** Meta-Controller for Adversarial Contextual Algorithm with Free Observation Budget

---

**Input:** Time horizon $T$, action set $\mathcal{A}$, policy space $\Pi$, observation budget $B$, cost $c$, hyperparameter $\gamma$
1: **if** $B \geq 1.5c^{-\frac{2}{3}}(|\mathcal{A}| \ln |\Pi|)^{\frac{1}{3}} T^{\frac{2}{3}}$ **then**
2:     Let $\eta_t = \frac{1}{T}\sqrt{\frac{B}{|\mathcal{A}| \ln |\Pi|}}, \forall t \in [T]$
3:     **Run Algorithm 4** $(T, \mathcal{A}, \{\eta_t\}_{t=1}^{T}, \Pi, B, c, \gamma)$
4: **else**
5:     Let $\eta_t = c^{-\frac{1}{3}}|\mathcal{A}|^{-\frac{1}{3}} \ln^{\frac{2}{3}}(|\Pi|)t^{-\frac{2}{3}}, \forall t \in [T]$
6:     **Run Algorithm 6** $(T, \mathcal{A}, \{\eta_t\}_{t=1}^{T}, \Pi, B, c, \gamma)$
7: **end if**

---

Theorem 3.2 summarizes the performance guarantees of the proposed meta-controller across different budget regimes. In the limited budget regime (first case), the regret bound consists of two terms: the first term $(c|\mathcal{A}| \ln |\Pi|)^{\frac{1}{3}} T^{\frac{2}{3}}$ represents the inherent learning complexity, while the second term $-cB$ reflects the benefit derived from utilizing the available budget. In the sufficient budget regime (second case), the regret scales as $\mathcal{O}\left(T\sqrt{|\mathcal{A}| \ln |\Pi|/B}\right)$, demonstrating that the algorithm effectively leverages the abundant observation resources to achieve improved performance. This bound exhibits the characteristic square-root dependence on the inverse budget that is typical for bandit problems with ample exploration resources.

The threshold between these two regimes occurs at $B = \Theta(c^{-\frac{2}{3}}(|\mathcal{A}| \ln |\Pi|)^{\frac{1}{3}}(\ln |\mathcal{A}|)^{-\frac{1}{3}} T^{\frac{2}{3}})$, which aligns with the theoretical lower bound established in Theorem 3.1. This consistency confirms that our meta-controller optimally adapts to the available budget.

**Theorem 3.2.** The upper bound of Algorithm 2 is given by

$$\mathfrak{R}(T) \leq \begin{cases} \mathcal{O}\left((c|\mathcal{A}| \ln |\Pi|)^{\frac{1}{3}} T^{\frac{2}{3}} - cB\right), \\ \quad \text{if } B = \mathcal{O}(c^{-\frac{2}{3}}(|\mathcal{A}| \ln |\Pi|)^{\frac{1}{3}} T^{\frac{2}{3}}) \\ \mathcal{O}\left(T\sqrt{\frac{|\mathcal{A}| \ln(|\Pi|)}{B}}\right), \\ \quad \text{if } B = \Omega(c^{-\frac{2}{3}}(|\mathcal{A}| \ln |\Pi|)^{\frac{1}{3}} T^{\frac{2}{3}}) \end{cases}$$

## 4. Stochastic Setting with Paid Observations

In previous sections, we examined adversarial paid observation without assumptions on outcomes or policy structure. This generality sacrificed efficiency, requiring enumeration over the entire (finite) policy space. A natural question is whether more efficient algorithms are possible in a stochastic setting with structural assumptions on policies.

Here we answer this positively via an oracle-efficient algorithm in a function approximation framework. Using an online oracle that minimizes empirical loss, our method remains practical and achieves strong performance. This allows us to incorporate structure—such as a policy space induced by a function class—and maintain computational efficiency even when the function space is infinite.

### 4.1. Function Approximation and Online Oracle

We assume that each loss function $\ell_t(\cdot) : \mathcal{A} \to [0, 1]$ is drawn independently from a fixed distribution $\mathbb{P}_{\ell_t}(\cdot|x_t)$, where $\mathbb{P}_{\ell_1}(\cdot), \cdots, \mathbb{P}_{\ell_T}(\cdot)$ and $x_1, \cdots, x_T$ are selected by an adaptive adversary. We then make a standard realizability assumption. Namely, we assume that the learner has access to a class of value functions $\mathcal{F} \subset (\mathcal{X} \times \mathcal{A} \to [0, 1])$(e.g., neural networks, kernels, or forests) that models the mean of the loss distribution (Foster & Rakhlin, 2020b;a).
**Assumption 4.1** (Realizability). There exists a function $f^* \in \mathcal{F}$ such that for all $t$, $f^*(x_t, a) = \mathbb{E}[\ell_t(a)|x_t]$.

In this setting, we consider the policies induced by $\mathcal{F}$, denoted by $\Pi$. For any regression function $f \in \mathcal{F}$, the induced policy is defined as $\pi_f(x) = \arg\min_{a \in \mathcal{A}} f(x, a)$. Given the definition of (1), the aim of the learner is to minimize their regret to the optimal policy $\pi^* \in \Pi$.

We define a benchmark policy $\pi_{f^*}$ and let $\Delta(T) = \sum_{t=1}^{T} \left[\ell_t(\pi_{f^*}(x_t)) - \ell_t(\pi^*(x_t))\right] \geq 0$. The regret of (1) can then be decomposed as $\mathfrak{R}(T) =$

$$\Delta(T) + \mathbb{E}\left[\underbrace{\sum_{t=1}^{T}(\ell_t(a_t) - \ell_t(\pi_{f^*}(x_t)))}_{\text{Learning Performance w.r.t. } \pi_{f^*}} + \underbrace{\sum_{t=1}^{T}\sum_{a \in O_t} c_t(a)}_{\text{Observation Cost}}\right].$$

**Proposition 4.1.** Suppose $|\mathcal{F}| < \infty$. Then $\Delta(T) \leq 2\sqrt{2T \log |\mathcal{F}|}$.

The above proposition shows that the suboptimality of the benchmark policy $\pi_{f^*}$ is bounded by $\mathcal{O}(\sqrt{T})$.

**Remark 4.1.** Note the following two points. First, our framework can accommodate misspecification while preserving computational efficiency, though we omit details for clarity. Second, Assumption 4.1 defines a "semi-adversarial" setting: contexts are chosen arbitrarily, while losses can be chosen arbitrarily subject only to their conditional mean.

We assume access to an online regression oracle $\mathbf{Alg}_{Sq}$ for function class $\mathcal{F}$, which performs online learning with squared loss. In each round $t \in [T]$, the oracle outputs an estimator $\hat{f}_t \in \mathcal{F}$ and then receives context–action–loss tuples $\{(x_t, a, \ell_t(a))\}_{a \in O_t}$ with $O_t \subset \mathcal{A}$. The oracle aims to accurately predict the loss given the context and action, and its performance is measured via the square loss—a standard setup in function approximation (Foster et al., 2018; Zhang et al., 2024). We quantify its cumulative performance by the square-loss regret relative to the best function in $\mathcal{F}$.

**Assumption 4.2** (Bounded square-loss regret). The regression oracle $\mathbf{Alg}_{Sq}$ guarantees that for any sequence $\{(x_t, a, \ell_t(a))\}_{a \in O_t, t \in [T]}$, $\sum_{t=1}^{T} \sum_{a \in O_t} (\hat{f}_t(x_t, a) - \ell_t(a))^2 - \inf_{f \in \mathcal{F}} \sum_{t=1}^{T} \sum_{a \in O_t} (f(x_t, a) - \ell_t(a))^2 \leq \mathbf{Reg}_{Sq}(T)$.

Typical function spaces admit known regret guarantees for the regression oracle. For finite $\mathcal{F}$, Vovk's aggregation algorithm achieves $\mathbf{Reg}_{Sq}(T) = \mathcal{O}(\log |\mathcal{F}|)$ (Vovk, 1995). For a $d$-dimensional linear function class, $\mathbf{Reg}_{Sq}(T) = \mathcal{O}(d \log(T/d))$ (Kalai & Vempala, 2002).

### 4.2. Oracle-Efficient Algorithm Design

We present our method in Algorithm 3. Upon receiving the context $x_t$ at each round $t$, the algorithm first computes $\hat{f}_t$ by utilizing the regression oracle. Subsequently, the basic sampling technique is employed based on the inverse gap weighting approach (Foster & Rakhlin, 2020a). This sampling probability is designed to achieve optimal regret in the cost-free setting. The observation probability $p_t(a)$ we design is almost identical to that presented in Algorithm 1. Finally, the algorithm feeds the observation set $O_t$ to the oracle $\mathbf{Alg}_{Sq}$ to update the estimator.

To elucidate the clear structure of regret, we introduce the following quantity: $Q(q_t, p_t; \hat{f}_t, x_t, \eta_t, \{c_t(a)\}_{a \in \mathcal{A}})$

$= \sup_{\substack{a^* \in \mathcal{A}, \\ f^* \in \mathcal{F}}} \mathbb{E}_{a \sim q_t, O_t \sim p_t} \Big[ f^*(x_t, a) - f^*(x_t, a^*) - \frac{\eta_t}{4} \sum_{a' \in O_t} (\hat{f}(x_t, a') - f^*(x_t, a'))^2 - \frac{\eta_t}{4} \sum_{a \in O_t} c_t(a) \Big]$.

This quantity, $Q(q_t, p_t; \hat{f}_t, x_t, \eta_t, \{c_t(a)\}_{a \in \mathcal{A}})$, is motivated by the decision-estimation coefficient (DEC) introduced in (Foster et al., 2021), incorporating the paid observations. Under the realizability assumption, we establish the following Lemma 4.1 which decomposes the regret into components based on this quantity.

**Lemma 4.1.** In Algorithm 3, we have $\mathfrak{R}(T) \leq \Delta(T) +$

---

**Algorithm 3** Contextual Algorithm with Paid Observations and Online Oracle

**Input:** time horizon $T$, action set $\mathcal{A}$, a regression oracle $\mathbf{Alg}_{Sq}$, hyperparameter $\gamma$
1: **for** epoch $t = 1, 2, \cdots, T$ **do**
2:    A loss vector $\{\ell_t(a)\}_{a \in \mathcal{A}}$ is generated by the adversary but not disclosed to the learner
3:    A context $x_t$ is generated by the adversary and revealed to the learner
4:    Observe the observation costs $c_t(a)$ for all $a \in \mathcal{A}$
5:    Obtain an estimator $\hat{f}_t$ from the online oracle $\mathbf{Alg}_{Sq}$
6:    Compute the sampling probability $q_t(a) = \begin{cases} \frac{1}{|\mathcal{A}| + \gamma(\hat{f}_t(x_t, a) - \hat{f}_t(x_t, a_t^*))}, a \neq a_t^*, \\ 1 - \sum_{a \neq a_t^*} q_t(a), a = a_t^*, \end{cases}$ where $a_t^* = \arg\min_{a \in \mathcal{A}} \hat{f}_t(x_t, a)$.
7:    Sample an action $a_t \sim q_t(a)$ and take the action
8:    Set $\eta_t = \left( \frac{\sum_{s=1}^{t} \sqrt{\sum_{a \in \mathcal{A}} c_s(a)}}{\mathbf{Reg}_{Sq}(T)} \right)^{\frac{2}{3}} + \sqrt{\frac{t}{\mathbf{Reg}_{Sq}(T)}},$
9:    Compute $p_t(a) = \min\left\{ 1, \sqrt{\frac{q_t(a)(1 - q_t(a))}{\eta_t c_t(a)}} \right\}$
10:   For each $a$, with probability $p_t(a)$, pay the cost $c_t(a)$ to observe a loss $\ell_t(a)$
11:   Update $\mathbf{Alg}_{Sq}$ with tuples $\{(x_t, a, \ell_t(a))\}_{a \in O_t}$
12: **end for**

---

$\sum_{t=1}^{T} Q(q_t, p_t; \hat{f}_t, x_t, \eta_t, \{c_t(a)\}_{a \in \mathcal{A}}) + \frac{\eta_T}{4} \mathbf{Reg}_{Sq}(T)$.

Theorem 4.1 establishes the regret upper bound through the following technical advances: (1) transforming expectations over action distributions into tractable summations and reducing functional suprema to parameter optimization via extremum conditions; (2) reformulating the minimax problem over the optimal action $a_t^*$ into a convex-concave optimization over auxiliary distributions, enabling order exchange using Sion's minimax theorem; (3) deriving closed-form expressions for the sampling distribution $p_t(a) = \min\left\{ 1, \sqrt{\frac{q_t(a)(1 - q_t(a))}{\eta_t c_t(a)}} \right\}$, adapting $q_t$ to cost constraints, and designing the multi-scale step size $\eta_t$ to control cumulative error growth.

**Theorem 4.1.** Let $\gamma = \sqrt{|\mathcal{A}| T / (\mathbf{Reg}_{Sq}(T) + \log(2\delta^{-1}))}$, the regret of Algorithm 3 satisfies $\mathfrak{R}(T) \leq \Delta(T) + 3(\mathbf{Reg}_{Sq}(T))^{\frac{1}{3}} \left( \sum_{t=1}^{T} \sqrt{\sum_{a \in \mathcal{A}} c_t(a)} \right)^{\frac{2}{3}} + 2\sqrt{T \mathbf{Reg}_{Sq}(T)}$ with probability at least $1 - \delta$.

Our algorithm and results can be easily extended to handle infinite functions using standard learning-theoretic tools such as metric entropy (Foster & Rakhlin, 2020a).

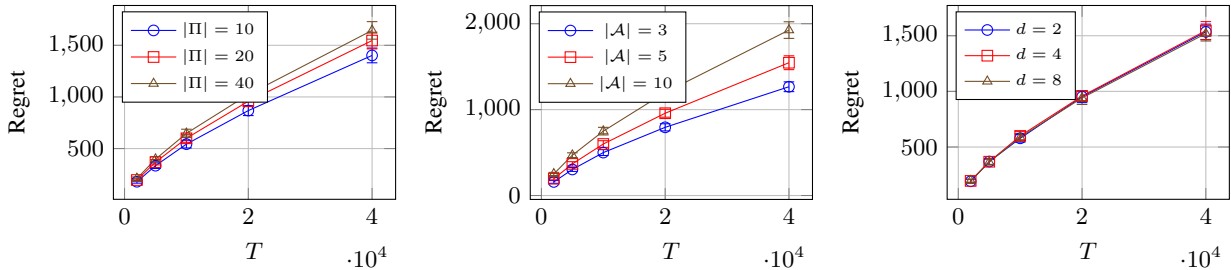

*Figure 2.* Cumulative regret of Algorithm 1 under different parameter settings.

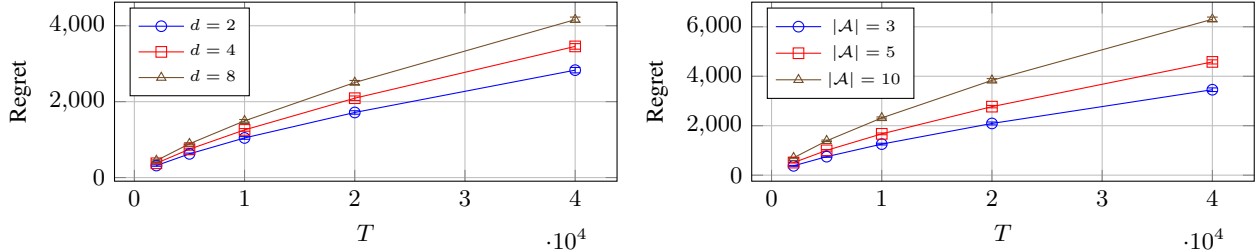

*Figure 3.* Cumulative regret of Algorithm 3 under different parameter settings.

## 5. Numerical Experiments

**Comparison with different parameter configuration.** A full-factorial parameter grid is constructed to cover typical scenarios in contextual bandit problems, with three key parameters varied across practical ranges. This design results are characterized as follows: the number of decision rounds ($T \in \{2000, 5000, 10000, 20000, 40000\}$), the context dimension ($d \in \{2, 4, 8\}$), and the size of the action space ($|\mathcal{A}| \in \{3, 5, 10\}$). For each configuration, a standardized environment is generated using: $L_2$-normalized Gaussian contexts (with mean 0 and variance 1), and observation costs drawn from $\mathcal{U}(0.01, 0.5)$. To account for stochastic variability, each parameter combination is evaluated over 50 independent replications.

For Algorithm 1, the policy class size $|\Pi|$ is set to $\{10, 20, 40\}$ to assess its influence on regret. All policies are derived from linear functions, with weights sampled from a standard normal distribution. Action-specific losses are generated from $\mathcal{U}(0, 1)$.

For Algorithm 3, we adopt the linear function class for regression. The linear parameters for each action are sampled from a standard normal distribution, and the losses are generated as the inner product of the context and action parameters, plus standard Gaussian noise.

Figure 2 shows how regret changes when varying individual parameters. In all cases, cumulative regret grows sublinearly with $T$. A larger policy space increases regret, as finding the optimal policy becomes harder with more candidates. Similarly, more actions lead to higher regret due to the increased exploration burden.

In contrast, the context dimension $d$ has negligible effect, with nearly identical regret curves for $d = 2, 4, 8$. This occurs because the policy class size $|\Pi|$, not the dimension, determines the learning complexity in this setting.

Figure 3 shows that the performance of Algorithm 3 follows trends similar to those of Algorithm 1. The key advantage of Algorithm 3 lies in its ability to handle very large—in fact, infinite—policy classes without explicit enumeration.

To assess the computational efficiency of Algorithm 3, we measure its average run time across different settings. With $(T, |\mathcal{A}|, d) = (40000, 10, 8)$, Table 1 confirms the clear efficiency advantage of Algorithm 3.

As the policy count $|\Pi|$ in Algorithm 1 grows from 10 to 40, its average run time increases from 33.2 s to 94.0 s. In contrast, Algorithm 3 maintains a consistently low average run time of 2.0 s and a small standard deviation, enabling faster and more predictable processing. These results demonstrate the efficiency of function approximation in contextual bandits with paid observations.

| Algorithm 1 | $|\Pi| = 10$ | $|\Pi| = 20$ | $|\Pi| = 40$ | Algorithm 3 |
|---|---|---|---|---|
| Time (Std) | 33.2 (1.3) | 48.2 (1.7) | 94.0 (3.4) | 2.0 (0.6) |

*Table 1.* execution time comparison (in seconds) between Algorithm 1 and Algorithm 3

**Comparison to baseline algorithms.** We compare our algorithms with several baselines: linUCB, a standard contextual bandit algorithm that does not account for observation costs,

and Tucker et al. (2023), the only existing work on linear contextual bandits with costly observations. The contextual feature dimension is $d = 10$, the number of arms is $|\mathcal{A}| = 5$, and the horizon is $T = 10,000$. Since Tucker et al. (2023) consider fixed costs, we set $c = 0.5$ for all algorithms. All experiments are repeated for 50 independent runs.

| Algorithm | Final Average Cumulative Regret |
|---|---|
| linUCB | $5003.32 \pm 1.04$ |
| (Tucker et al., 2023) | $3697.13 \pm 26.80$ |
| Algorithm 3 | $2899.75 \pm 48.80$ |

*Table 2.* Final Average Cumulative Regret of Different Algorithms.

In Table 2, linUCB incurs the highest cost-inclusive regret because it does not account for observation costs in its decision-making, resulting in excessive spending on observations; its low variance reflects this deterministic observation strategy. Tucker's algorithm achieves lower regret by incorporating cost awareness, but is outperformed by Algorithm 3. Our method obtains the lowest regret by jointly optimizing the exploration-exploitation and cost-information trade-offs. The slightly higher variance of Algorithm 3 reflects its adaptive observation strategy, which introduces more variability but achieves better mean performance.

## 6. Conclusion

This paper studies contextual bandits with paid observations. We propose minimax optimal algorithms and empirically validate them in Section 5. We also connect to existing work, providing a unified view of related topics.

Several future directions remain open. First, extending the analysis to heterogeneous observation costs in the free budget setting is an important step: while our lower bound and meta-controller currently assume a uniform cost $c$ across actions and rounds, many practical scenarios involve non-uniform or time-varying costs, and handling such general cost structures would require a more nuanced exploration-cost trade-off. Extensions to non-stationary or combinatorial action spaces could broaden applicability. Practically, integrating other oracle-efficient approach in fully adversarial settings and conducting large-scale experiments are promising. Finally, links to multi-agent systems and differential privacy offer rich avenues for further work.

## Impact Statement

This paper presents work whose goal is to advance the field of machine learning. There are many potential societal consequences of our work, none of which we feel must be specifically highlighted here.

## Acknowledgement

This work is generously supported by the National Natural Science Foundation of China (NSFC) under Grant No. 72501238 and by the Hong Kong Research Grants Council (UGC) through the Theme-based Research Project T32-615/24-R and the General Research Fund No. 16500023.

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

# A. Motivating Examples

The following examples show the **contextual bandit with paid observations** framework, where agents must decide whether to pay a cost to observe a reward. Each scenario is defined by its **context**, **actions**, **observation cost**, **reward**, and **goal**.

## 1. IoT Device Power Management

- **Context**: Battery level, environmental conditions (e.g., temperature, noise).

- **Actions**: Activate high-power sensor (e.g., camera); Use low-power sensor (e.g., motion detector); Enter sleep mode.

- **Observation Cost**: Energy consumed to measure and transmit reward data.

- **Reward**: Accuracy or usefulness of sensor readings.

- **Goal**: Maximize data quality while minimizing energy consumption.

## 2. Chatbot Optimization

- **Context**:
  - User query and conversation history.
  - User profile (if available).

- **Actions**: Select a response (utterance) from a predefined set.

- **Observation Cost**: Human evaluation cost (time/money per rating).

- **Reward**: User satisfaction score (e.g., thumbs-up/down, rating scale).

- **Goal**: Maximize user satisfaction while minimizing feedback costs.

## 3. Holistic Recommendation Systems

- **Context**:
  - User demographics and past engagement.
  - Content features (e.g., topic, video length).

- **Actions**: Recommend content (e.g., articles, videos).

- **Observation Cost**: Hiring human assessors for long-term feedback.

- **Reward**: Long-term user satisfaction (e.g., well-being, retention).

- **Goal**: Optimize holistic satisfaction while minimizing assessment costs.

## 4. Medical Insurance Service Contracts

- **Context**:
  - Hospital performance history and contract terms.
  - Regional healthcare demands.

- **Actions**:
  - Sign contract with a hospital.
  - Decide whether to inspect service quality.

- **Observation Cost**: Inspection fee (e.g., independent auditor).

- **Reward**: Service quality score (if inspected) or estimated quality.

- **Goal**: Maximize service quality while minimizing inspection costs.

# B. Discussion

**Comparison to (Seldin et al., 2014).** Seldin et al. (2014) first studied the problem of multiarmed bandits with paid observations and provided algorithms and matching (up to logarithmic factors) upper and lower bounds. Our algorithm extend their work to the more general contextual bandit setting, where the loss functions are associated with a context and the policy space can be infinite. We introduce an oracle-efficient algorithm, addressing the computational inefficiency of their approach for large policy spaces.

**Comparison to (Tucker et al., 2023).** In contrast to (Tucker et al., 2023), which primarily focuses on bandit problems with costly reward observations in a fully stochastic setting, we address the challenges of the adversarial contextual bandit problem with a general policy space. While (Tucker et al., 2023) proposes an algorithm for the multi-armed bandits, incurring regrets of $\mathcal{O}(c^{1/3}T^{2/3})$, they do not provide theoretical guarantees in terms of regret upper or lower bounds for their linear contextual bandit algorithm. We bridge this research gap by considering the both stochastic and adversarial contextual bandits with a general policy space, providing both upper and lower bounds on the regret. Specifically, from previous discussion, $\mathcal{O}((cd|\mathcal{A}|\log T)^{\frac{1}{3}}T^{\frac{2}{3}})$ is achievable in linear contextual cases. Our approaches handle the complexities introduced by the costly observations and offer meaningful theoretical guarantees.

**Comparison to (Avner et al., 2012).** Their decoupled exploration setting fixes the amount of information to one reward observation per round. In contrast, our framework allows the learner to select an arbitrary subset of arms to observe at a cost that may vary across arms and over time. This additional flexibility introduces the fundamental cost-information trade-off that drives the phase transition in regret rate. Moreover, our analysis handles adversarial contexts and general policy classes, whereas Avner et al. (2012) focus on the non-contextual setting.

**Comparison to (Thune & Seldin, 2018).** Their setting provides exactly one free extra observation per round in a non-contextual prediction problem; SODA achieves regret $O(\epsilon\sqrt{|\mathcal{A}|T\ln|\mathcal{A}|})$ against adversarial losses with range $\epsilon$. While their setting resembles ours when $B = 2T$, the results are not directly comparable due to different assumptions on observation structure and loss range.

**Budgeted MAB.** The research conducted by (Amin et al., 2015) explores a budgeted variant of the MAB problem involving learning from expert advice with $N$ experts. In this scenario, each interaction with an expert incurs a cost denoted as $C$, while a predetermined budget $B_0$ is imposed to restrict the total cost of consultations during each prediction round. To address this challenge, (Amin et al., 2015) propose an online learning algorithm that guarantees a $\mathcal{O}(\sqrt{\frac{C}{B_0}T\ln N})$ regret bound after $T$ rounds. Interestingly, by defining $M = \frac{B_0}{C}$ and $\Pi = [N]$, it becomes apparent that $MT$ signifies the overall number of observations within their framework. Since the cost incurred by consultations is not factored into the regret, our Lemma 2.2 can be applied and refines their existing lower bound of $\Omega(\sqrt{\frac{C}{B_0}T})$ by incorporating an additional term of $\sqrt{\ln|\Pi|}$ in the contextual bandit setting. To establish connections to this topic, we propose Algorithm 8 in Section E which deals with the adversarial contextual bandit with time-varying budgets $M_t$. The following Corollary B.1, combined with the lower bound in Lemma 2.2, indicates the optimality of our algorithm.

**Corollary B.1.** If we choose $\eta_t = \sqrt{\ln|\Pi|/(|\mathcal{A}|\sum_{s=1}^{t}\frac{1}{M_s})}$, the regret of Algorithm 8 satisfies $\mathfrak{R}(T) \leq \mathcal{O}\left(\sqrt{|\mathcal{A}|\ln|\Pi|\sum_{t=1}^{T}\frac{1}{M_t}}\right)$.

*Proof.* It is straightforward to prove that $\hat{\ell}_t(a)$ is an unbiased estimator for $\ell_t(a)$, because we uniformly sample $M_t$ actions on $\mathcal{A}$ without replacement. Since Algorithm 8 follows the template of Algorithm 1, the proof of Lemma C.1 and Lemma 2.1 show that

$$\mathfrak{R}(T) \leq \frac{\ln|\Pi|}{\eta_{T+1}} + \sum_{t=1}^{T}\eta_t\sum_{a\in\mathcal{A}}\frac{|\mathcal{A}|q_t(a)}{M_t}.$$

If we choose $\eta_t = \sqrt{\frac{\ln|\Pi|}{|\mathcal{A}|\sum_{s=1}^{t}M_s^{-1}}}$ and use the same induction trick in the proof of Theorem 2.1, we will have

$$\mathfrak{R}(T) \leq \mathcal{O}\left(\sqrt{|\mathcal{A}|\ln|\Pi|\sum_{t=1}^{T}\frac{1}{M_t}}\right).$$

$\square$

**Uninformed costs.** In the previous setting, we consider the scenario with informed costs. However, in reality, the uninformed costs are common. Here, the learner remains unaware of the costs associated with different actions until after making decisions. Since the learner has no prior knowledge of the costs, we need to make a distributional assumption regarding the costs. Specifically, we assume that each cost is independently and identically sampled from a sub-Gaussian distribution with a mean of $\bar{c}(a)$. In light of this assumption, we modify the targeted regret as follows: $\Re(T) = \sum_{t=1}^{T} \left[ \ell_t(a_t) - \ell_t(\pi^*(x_t)) + \sum_{a \in O_t} \bar{c}(a) \right]$.

The core concept behind our approach is to utilize upper confidence bounds. We leverage the observed costs for each action to compute the empirical mean, denoted as $\hat{c}_t(a) = \frac{1}{t} \sum_{s=1}^{t} c_s(a)$, and construct the upper confidence bound (UCB) as follows:

$$U_t(a) = \hat{c}_t(a) + \sqrt{\frac{2 \log(4|\mathcal{A}|T/\delta)}{t}}. \tag{2}$$

This construction ensures that, with high probability, the true mean cost $\bar{c}(a)$ is lower than the UCB $U_t(a)$. By replacing the costs $c_t(a)$ with $U_t(a)$, we derive the following corollary.

**Corollary B.2.** Let $\gamma = \sqrt{|\mathcal{A}|T/(\mathbf{Reg}_{Sq}(T) + \log(4\delta^{-1}))}$. The regret of Algorithm 7 with the uninformed costs is upper

bounded by $2(\mathbf{Reg}_{Sq}(T))^{\frac{1}{3}} T^{\frac{2}{3}} \left( \sqrt{\sum_{a \in \mathcal{A}} \bar{c}(a)} \right)^{\frac{2}{3}} + 3.2(\mathbf{Reg}_{Sq}(T))^{\frac{1}{3}} |\mathcal{A}|^{\frac{1}{3}} T^{\frac{1}{2}} (\log(4|\mathcal{A}|T/\delta))^{\frac{1}{6}} + \sqrt{T \mathbf{Reg}_{Sq}(T)}/4 + \Delta(T)$ with probability at least $1 - \delta$.

The uninformed costs only increase additional regret $\tilde{\mathcal{O}}(\sqrt{T})$, which is negligible compared with $\mathcal{O}(T^{2/3})$.

**Metric entropy.** Suppose $\mathcal{F}$ is equipped with a maximum norm $\|\cdot\|$. We can consider an $\epsilon$-covering $\mathcal{F}_\epsilon$ of $\mathcal{F}$ under maximum norm. Since $|\mathcal{F}_\epsilon|$ is finite, we can directly replace $\mathcal{F}$ with $\mathcal{F}_\epsilon$ and do not change any algorithmic procedure. Thanks to the property of $\epsilon$-covering, there exists a function $f_\epsilon^* \in \mathcal{F}_\epsilon$ such that $\|f_\epsilon^* - f^*\| \leq \epsilon$. The term $\log|\mathcal{F}_\epsilon|$ is the so-called metric entropy. For example, if $\mathcal{F}$ is $d$-dimensional, then $\log|\mathcal{F}_\epsilon| \sim d \log(\frac{1}{\epsilon})$. If we take $\epsilon = 1/T$, then $\Delta(T) \leq 2\sqrt{2dT}$ and thus $\Re(T) \leq 3(cd|\mathcal{A}| \log T)^{\frac{1}{3}} T^{\frac{2}{3}} + 6\sqrt{dT \log T} + 1$.

# C. Proofs for Section 2

## C.1. Proof of Lemma C.1

The significance of Lemma 1 is that it expresses regret in terms of estimators rather than true losses. By applying the policy from Algorithm 1 and the optimal policy, we directly relate the left-hand side to the regret of Algorithm 1. This allows us to use the known loss estimator instead of the unknown true loss in our regret analysis.

For notation brevity, we use $\mathbb{E}_t[\cdot] = \mathbb{E}[\cdot | \mathcal{H}_{t-1}]$. The definitions of $\mathcal{H}_t$ has been specified in the sections of settings.

**Lemma C.1.** In Algorithm 1, we have $\mathbb{E}\left[\sum_{a \in \mathcal{A}} (q_t(a) - \mathbb{I}\{\pi^*(x_t) = a\})\hat{\ell}_t(a)\right] = \mathbb{E}\left[\sum_{a \in \mathcal{A}} (q_t(a) - \mathbb{I}\{\pi^*(x_t) = a\})\ell_t(a)\right]$.

*Proof.* We first prove that $\mathbb{E}_t[\hat{\ell}_t(a)]$ is an unbiased estimator of $\ell_t(a)$. We have that

$$\mathbb{E}_t[\hat{\ell}_t(a)] = \mathbb{E}_t\left[\mathbb{I}\{a \in O_t\}\frac{\ell_t(a)}{p_t(a)}\right]$$

$$= \mathbb{E}_t[\mathbb{I}\{a \in O_t\}]\frac{\ell_t(a)}{p_t(a)}$$

$$= \ell_t(a).$$

Then we can prove the following equality.

$$\mathbb{E}\left[\sum_{a \in \mathcal{A}} (q_t(a) - \mathbb{I}\{\pi^*(x_t) = a\})\hat{\ell}_t(a)\right]$$

$$= \mathbb{E}\left[\sum_{a \in \mathcal{A}} (q_t(a) - \mathbb{I}\{\pi^*(x_t) = a\})\mathbb{E}_t[\hat{\ell}_t(a)]\right]$$

$$= \mathbb{E}\left[\sum_{a \in \mathcal{A}} (q_t(a) - \mathbb{I}\{\pi^*(x_t) = a\})\ell_t(a)\right].$$

The second equality is due to the rule of the total expectation. $\qquad\square$

## C.2. Proof of Lemma 2.1

**Lemma C.2.** If we choose $\gamma \in (0,1)$ and $\eta_t$ is a non-increasing sequence, then the following inequality holds $\mathbb{E}\left[\sum_{t=1}^T \sum_{a \in \mathcal{A}} (q_t(a) - \mathbb{I}\{\pi^*(x_t) = a\})\ell_t(a)\right] \leq \frac{(1-\gamma)\ln|\Pi|}{\eta_{T+1}} + \mathbb{E}\left[\sum_{t=1}^T \eta_t \sum_{a \in \mathcal{A}} \frac{q_t(a)}{p_t(a)}\right] + 2\gamma T$.

*Proof.* The following result holds for each $t \in [T]$,

$$ln\left(\frac{\sum_{\pi \in \Pi} w_t(\pi) \cdot e^{-\eta_t \hat{\ell}_t(\pi(x_t))}}{\sum_{\pi \in \Pi} w_t(\pi)}\right)$$

$$= ln\left(\frac{\sum_{a \in \mathcal{A}} \sum_{\pi \in \Pi: \pi(x_t)=a} w_t(\pi) \cdot e^{-\eta_t \hat{\ell}_t(a)}}{\sum_{\pi \in \Pi} w_t(\pi)}\right)$$

$$= ln\left(\sum_{a \in \mathcal{A}} \frac{q_t(a) - \gamma/|\mathcal{A}|}{1-\gamma} e^{-\eta_t \hat{\ell}_t(a)}\right)$$

$$\leq ln\left(\sum_{a \in \mathcal{A}} \frac{q_t(a) - \gamma/|\mathcal{A}|}{1-\gamma}(1 - \eta_t \hat{\ell}_t(a) + \eta_t^2(\hat{\ell}_t(a))^2)\right)$$

$$= ln\left(1 + \sum_{a \in \mathcal{A}} \frac{q_t(a) - \gamma/|\mathcal{A}|}{1-\gamma}(-\eta_t \hat{\ell}_t(a) + \eta_t^2(\hat{\ell}_t(a))^2)\right)$$

$$\leq \frac{1}{1-\gamma} \sum_{a \in \mathcal{A}} q_t(a)\left(-\eta_t \hat{\ell}_t(a) + \eta_t^2(\hat{\ell}_t(a))^2\right) + \frac{\eta_t \gamma}{|\mathcal{A}|} \sum_{a \in \mathcal{A}} \hat{\ell}_t(a),$$

where the first inequality is due to $e^{-z} \leq 1 - z + z^2$ for $z \geq 0$ and the second inequality is due to $\ln(1+z) \leq z$ for $z > -1$. Reordering the above term yields

$$\sum_{a \in \mathcal{A}} q_t(a) \hat{\ell}_t(a) \leq -\frac{1-\gamma}{\eta_t} ln \left( \frac{\sum_{\pi \in \Pi} w_t(\pi) \cdot e^{-\eta_t \hat{\ell}_t(\pi(x_t))}}{\sum_{\pi \in \Pi} w_t(\pi)} \right) + \eta_t \sum_{a \in \mathcal{A}} q_t(a)(\hat{\ell}_t(a))^2 + \frac{\gamma}{|\mathcal{A}|} \sum_{a \in \mathcal{A}} \hat{\ell}_t(a).$$

Now we need to lower bound $\frac{1}{\eta_t} ln \left( \frac{\sum_{\pi \in \Pi} w_t(\pi) \cdot e^{-\eta_t \hat{\ell}_t(\pi(x_t))}}{\sum_{\pi \in \Pi} w_t(\pi)} \right)$ for each $t$, which leads to the following steps. Notice that

$$\frac{1}{|\Pi|} \sum_{\pi \in \Pi} w_{t+1}(\pi) = \sum_{\pi \in \Pi} \frac{1}{|\Pi|} \exp \left( -\eta_{t+1} \sum_{s=1}^{t} \hat{\ell}_s(\pi(x_s)) \right)$$

$$= \sum_{\pi \in \Pi} \frac{1}{|\Pi|} \left( \exp \left( -\eta_t \sum_{s=1}^{t} \hat{\ell}_s(\pi(x_s)) \right) \right)^{\eta_{t+1}/\eta_t}$$

$$\leq \left( \frac{1}{|\Pi|} \sum_{\pi \in \Pi} \exp \left( -\eta_t \sum_{s=1}^{t} \hat{\ell}_s(\pi(x_s)) \right) \right)^{\eta_{t+1}/\eta_t}$$

$$= \left( \frac{1}{|\Pi|} \sum_{\pi \in \Pi} w_t(\pi) \cdot e^{-\eta_t \hat{\ell}_t(\pi(x_t))} \right)^{\eta_{t+1}/\eta_t}.$$

The inequality is Jensen's inequality for the concave function $z^{\eta_{t+1}/\eta_t}$ as $\eta_t$ is a decreasing sequence. Taking the logarithmic on both sides, we have that

$$\frac{1}{\eta_{t+1}} \ln \left( \frac{1}{|\Pi|} \sum_{\pi \in \Pi} w_{t+1}(\pi) \right) \leq \frac{1}{\eta_t} \ln \left( \frac{1}{|\Pi|} \sum_{\pi \in \Pi} w_t(\pi) \cdot e^{-\eta_t \hat{\ell}_t(\pi(x_t))} \right).$$

The above inequality implies that

$$\frac{1}{\eta_{t+1}} \ln \left( \frac{1}{|\Pi|} \sum_{\pi \in \Pi} w_{t+1}(\pi) \right) - \frac{1}{\eta_t} \ln \left( \frac{1}{|\Pi|} \sum_{\pi \in \Pi} w_t(\pi) \right) \leq \frac{1}{\eta_t} ln \left( \frac{\sum_{\pi \in \Pi} w_t(\pi) \cdot e^{-\eta_t \hat{\ell}_t(\pi(x_t))}}{\sum_{\pi \in \Pi} w_t(\pi)} \right).$$

Therefore, we have

$$\sum_{t=1}^{T} \frac{1}{\eta_t} ln \left( \frac{\sum_{\pi \in \Pi} w_t(\pi) \cdot e^{-\eta_t \hat{\ell}_t(\pi(x_t))}}{\sum_{\pi \in \Pi} w_t(\pi)} \right)$$

$$\geq \sum_{t=1}^{T} \frac{1}{\eta_{t+1}} \ln \left( \frac{1}{|\Pi|} \sum_{\pi \in \Pi} w_{t+1}(\pi) \right) - \frac{1}{\eta_t} \ln \left( \frac{1}{|\Pi|} \sum_{\pi \in \Pi} w_t(\pi) \right)$$

$$= \frac{1}{\eta_{T+1}} \ln \left( \sum_{\pi \in \Pi} w_{T+1}(\pi) \right) - \frac{\ln |\Pi|}{\eta_{T+1}}$$

$$\geq \frac{1}{\eta_{T+1}} \ln \left( w_{T+1}(\pi^*) \right) - \frac{\ln |\Pi|}{\eta_{T+1}}$$

$$= -\sum_{t=1}^{T} \hat{\ell}_t(\pi^*(x_t)) - \frac{\ln |\Pi|}{\eta_{T+1}}$$

$$= -\sum_{t=1}^{T} \sum_{a \in \mathcal{A}} \mathbb{I}\{\pi^*(x_t) = a\} \hat{\ell}_t(a) - \frac{\ln |\Pi|}{\eta_{T+1}}.$$

Combing the two bounds together, we have

$$
\mathbb{E}\left[\sum_{t=1}^{T}\sum_{a\in\mathcal{A}}(q_t(a) - \mathbb{I}\{\pi^*(x_t) = a\})\hat{\ell}_t(a)\right]
$$

$$
\leq \frac{(1-\gamma)\ln|\Pi|}{\eta_{T+1}} + \mathbb{E}\left[\sum_{t=1}^{T}\eta_t\sum_{a\in\mathcal{A}}q_t(a)(\hat{\ell}_t(a))^2\right] + \gamma\mathbb{E}\left[\sum_{t=1}^{T}\sum_{a\in\mathcal{A}}\left(\frac{1}{|\mathcal{A}|} - \mathbb{I}\{\pi^*(x_t) = a\}\right)\hat{\ell}_t(a)\right]
$$

$$
\leq \frac{(1-\gamma)\ln|\Pi|}{\eta_{T+1}} + \mathbb{E}\left[\sum_{t=1}^{T}\eta_t\sum_{a\in\mathcal{A}}q_t(a)(\hat{\ell}_t(a))^2\right] + 2\gamma T.
$$

Thanks to Lemma C.1, we have

$$
\mathbb{E}\left[\sum_{t=1}^{T}\sum_{a\in\mathcal{A}}(q_t(a) - \mathbb{I}\{\pi^*(x_t) = a\})\ell_t(a)\right]
$$

$$
= \mathbb{E}\left[\sum_{t=1}^{T}\sum_{a\in\mathcal{A}}(q_t(a) - \mathbb{I}\{\pi^*(x_t) = a\})\hat{\ell}_t(a)\right]
$$

$$
\leq \frac{(1-\gamma)\ln|\Pi|}{\eta_{T+1}} + \mathbb{E}\left[\sum_{t=1}^{T}\eta_t\sum_{a\in\mathcal{A}}q_t(a)(\hat{\ell}_t(a))^2\right] + 2\gamma T.
$$

It remains to bound the term $\mathbb{E}\left[\sum_{t=1}^{T}\eta_t\sum_{a\in\mathcal{A}}q_t(a)(\hat{\ell}_t(a))^2\right]$. We have

$$
\mathbb{E}\left[\sum_{t=1}^{T}\eta_t\sum_{a\in\mathcal{A}}q_t(a)(\hat{\ell}_t(a))^2\right] \leq \sum_{t=1}^{T}\eta_t\sum_{a\in\mathcal{A}}\frac{q_t(a)}{p_t^2(a)}\mathbb{E}_t[\mathbb{I}_t(a)] = \sum_{t=1}^{T}\eta_t\sum_{a\in\mathcal{A}}\frac{q_t(a)}{p_t(a)}.
$$

We conclude the proof by plugging all above inequalities.

$\square$

## C.3. Proof of Theorem 2.1

Given Lemma C.1, we are ready to prove Theorem 2.1.

*Proof.* It remains to find an expression for

$$
\sum_{t=1}^{T}\sum_{a\in O_t}c_t(a).
$$

From the algorithmic construction, we have

$$
\mathbb{E}\left[\sum_{t=1}^{T}\sum_{a\in O_t}c_t(a)\right] = \sum_{t=1}^{T}\sum_{a\in\mathcal{A}}c_t(a)p_t(a).
$$

Therefore, we have

$$
\mathfrak{R}(T) \leq \frac{\ln|\Pi|}{\eta_{T+1}} + \sum_{t=1}^{T}\sum_{a\in\mathcal{A}}\left[\frac{\eta_t q_t(a)}{p_t(a)} + c_t(a)p_t(a)\right].
$$

Now it suffices to consider the following optimization problem

$$
\min \frac{\eta_t q_t(a)}{p_t(a)} + c_t(a)p_t(a)
$$
$$
s.t. \quad 0 \leq p_t(a) \leq 1,
$$

which attains the minimum at $p_t^*(a) = \min\left\{1, \sqrt{\frac{\eta_t q_t(a)}{c_t(a)}}\right\}$.

If $p_t^*(a) = 1$, then $c_t(a) \leq \eta_t q_t(a)$, so

$$c_t(a) = \sqrt{c_t(a)} \cdot \sqrt{c_t(a)} \leq \sqrt{\eta_t q_t(a) c_t(a)}$$

and thus

$$\sum_{a \in \mathcal{A}: p_t^*(a)=1} \frac{\eta_t q_t(a)}{p_t(a)} + c_t(a) p_t(a) \leq \eta_t + \sum_{a \in \mathcal{A}} \sqrt{\eta_t q_t(a) c_t(a)}.$$

If $p_t^*(a) = \sqrt{\frac{\eta_t q_t(a)}{c_t(a)}}$, then we have

$$\frac{\eta_t q_t(a)}{p_t(a)} + c_t(a) p_t(a) = 2\sqrt{\eta_t q_t(a) c_t(a)}.$$

We combine the two cases together and obtain

$$\mathfrak{R}(T) \leq \frac{\ln|\Pi|}{\eta_{T+1}} + \sum_{t=1}^{T} \eta_t + 2 \sum_{t=1}^{T} \sum_{a \in \mathcal{A}} \sqrt{\eta_t q_t(a) c_t(a)}.$$

We finish the last step by choosing

$$\eta_t = \frac{1}{\left(\frac{\sum_{s=1}^{t}\sqrt{\sum_{a \in \mathcal{A}} c_s(a)}}{\ln|\Pi|}\right)^{\frac{2}{3}} + \sqrt{\frac{t}{\ln|\Pi|}}}, \eta_{T+1} = \eta_T,$$

as $\eta_t$ is non-increasing. We will bound the three terms in the following.

For the first term, we have

$$\frac{\ln|\Pi|}{\eta_{T+1}} = (\ln|\Pi|)^{\frac{1}{3}} \left(\sum_{s=1}^{T}\sqrt{\sum_{a \in \mathcal{A}} c_s(a)}\right)^{\frac{2}{3}} + \sqrt{T \ln|\Pi|}.$$

For the second term, we have

$$\sum_{t=1}^{T} \eta_t \leq \sum_{t=1}^{T} \sqrt{\frac{\ln|\Pi|}{t}} \leq 2\sqrt{T \ln|\Pi|}.$$

Fo the third term, using Jensen's inequality, we have

$$\sum_{a \in \mathcal{A}} \sqrt{q_t(a) c_t(a)} = \sum_{a \in \mathcal{A}} q_t(a) \sqrt{\frac{c_t(a)}{q_t(a)}} \leq \sqrt{\sum_{a \in \mathcal{A}} c_t(a)},$$

and we obtain that

$$\sum_{t=1}^{T} \sum_{a \in \mathcal{A}} \sqrt{\eta_t q_t(a) c_t(a)} \leq (\ln|\Pi|)^{\frac{1}{3}} \sum_{t=1}^{T} \frac{\sum_{a \in \mathcal{A}} \sqrt{q_t(a) c_t(a)}}{\left(\sum_{s=1}^{t}\sqrt{\sum_{a \in \mathcal{A}} c_s(a)}\right)^{\frac{1}{3}}}$$

$$\leq (\ln|\Pi|)^{\frac{1}{3}} \sum_{t=1}^{T} \frac{\sqrt{\sum_{a \in \mathcal{A}} c_t(a)}}{\left(\sum_{s=1}^{t}\sqrt{\sum_{a \in \mathcal{A}} c_s(a)}\right)^{\frac{1}{3}}}$$

$$\leq 1.5(\ln|\Pi|)^{\frac{1}{3}} \left(\sum_{t=1}^{T}\sqrt{\sum_{a \in \mathcal{A}} c_t(a)}\right)^{\frac{2}{3}},$$

where the last inequality comes from Lemma F.1.

Using Cauchy-Schwarz inequality, we have

$$\sum_{a \in \mathcal{A}} \sqrt{c_t(a) q_t(a)} \leq \sqrt{\sum_{a \in \mathcal{A}} c_t(a)},$$

and thus

$$\sum_{t=1}^{T} \sum_{a \in \mathcal{A}} \sqrt{\eta_t q_t(a) c_t(a)} \leq 1.5 (\ln |\Pi|)^{\frac{1}{3}} \left( \sum_{t=1}^{T} \sqrt{\sum_{a \in \mathcal{A}} c_t(a)} \right)^{\frac{2}{3}}.$$

Combining all the above terms, we have

$$\mathfrak{R}(T) \leq 4 (\ln |\Pi|)^{\frac{1}{3}} \left( \sum_{t=1}^{T} \sqrt{\sum_{a \in \mathcal{A}} c_t(a)} \right)^{\frac{2}{3}} + 4\sqrt{T \ln |\Pi|}.$$

As a corollary, if $c_t(a) = c$, then we will obtain

$$\mathfrak{R}(T) \leq \mathcal{O}\left( (c|\mathcal{A}| \ln |\Pi|)^{\frac{1}{3}} T^{\frac{2}{3}} + \sqrt{T \ln |\Pi|} \right).$$

$\square$

### C.4. Proof of Lemma 2.2

*Proof.* The basic idea is to reduce the original problem to a multi-armed bandit with finite contexts. We allow the adversary to generate losses from fixed distributions and show that there exist loss and context sequences satisfying the requirement. To construct bandit instances, we introduce a hyperparameter $\epsilon$ and a policy space $\Pi$ to be determined later. We further define $N = \lceil \ln |\Pi| / \ln |\mathcal{A}| \rceil$ as the number of contexts. Therefore, the original problem can be viewed as $N$ independent MAB problems.

**Policy space.** To link the losses and the policy space, we construct a function space $\mathcal{F}$ to induce a policy space. We begin with the policy class $\mathcal{G}$ consisting of all the mappings of the form $g : \mathcal{X} \to \mathcal{A}$, where $\mathcal{X} = [N]$. Such function $g$ maps a context to an action. It can be computed that the number of functions in $\mathcal{G}$ is equal to $|\mathcal{A}|^N \leq |\Pi|$. Each mapping $g \in \mathcal{G}$ defines a function $f_g \in \mathcal{F}$ as following

$$f_g(x, a) = \begin{cases} \frac{1}{2} - \epsilon, & \text{if } a = g(x) \\ \frac{1}{2}, & \text{otherwise.} \end{cases}$$

The function class $\mathcal{F}$ induces a policy space denoted as

$$\Pi = \{ \pi_f | \pi_f(x) = \arg\min_{a \in \mathcal{A}} f(x, a), \forall f \in \mathcal{F} \}.$$

Given the context $x$, the optimal action for $f_g(x, a)$ is $g(x)$, and other actions have suboptimal gap $\epsilon$.

**Loss.** The losses are generated by picking a function $f \in \mathcal{F}$ uniformly at random at the beginning. Equivalently, we choose a mapping $g$ that independently maps each context $x \in \mathcal{X}$ to a random action $a \in \mathcal{A}$, and set $f = f_g$. In each round $t$, a context $x_t$ is picked uniformly from $\mathcal{X}$. For any action $a$, a loss $\ell_t(a)$ is generated as a Bernoulli trial with probability $f(x, a)$. Let $\mathbb{P}'$ denote the loss distribution where the losses of any action $a$ for context $x$ are chosen to be $\{0, 1\}$ uniformly at random (the losses for other contexts $x' \neq x$ are still chosen according to $f(x', a)$), and let $\mathbb{E}'$ denote the expectation under $\mathbb{P}'$.

We condition on the choices of the contexts $x_t$, all the randomness of the algorithm **A**, and the values of $g(x')$ for $x' \neq x$. Thus the only randomness left is in the choice of $g(x)$, the realization of the losses in each round. When the context $x$ is fixed, the contextual bandit becomes an multi-armed bandit and is independent with other MAB instances. For fixed $x$, our construction is exactly the same as that in (Lattimore & Szepesvári, 2020) for MAB instances. Denote $T_x$ as the rounds

where the context $x_t$ is $x$. Let $S_{a|x}$ be a random variable denoting the number of choosing $a_t = a$ when $x_t = x$. Given the context $x$, we denote the total number of observations as $M_x|T_x|$, so we have $\sum_{x \in \mathcal{X}} M_x|T_x| = MT$.

Specifically, the notation $S_{g(x)|x}$ counts the number of choosing optimal actions when context $x$ occurs. The Pinsker's inequality shows

$$\mathbb{E}_{g(x)}[S_{g(x)|x}] \leq \mathbb{E}'[S_{g(x)|x}] + |T_x|\sqrt{2\epsilon^2 \mathbb{E}'[S_{g(x)|x}]}.$$

Futhermore, we have

$$\begin{aligned}
\mathbb{E}'[S_{g(x)|x}] \leq & \mathbb{E}'\left[\sum_{t \in T_x} \mathbb{I}\{g(x) \in O_t\}\right] \\
= & \sum_{t \in T_x} \mathbb{E}'[\mathbb{I}\{g(x) \in O_t\}] \\
= & \sum_{t \in T_x} \mathbb{E}'[|O_t|/|\mathcal{A}|] \\
= & M_x|T_x|/|\mathcal{A}|.
\end{aligned}$$

The third equality follows because $g(x)$ is independent of the choices of the contexts $x_t$, and $g(x')$ for $x' = x$, and its distribution is uniform on $\mathcal{A}$.

Notice that $\mathbb{E}_{g(x)}[\mathbb{E}[S_{g(x)|x}]]$ counts the number of rounds that **A** chooses the optimal action for $x$. We know that the regret given $x$ is at least

$$\epsilon(|T_x| - \mathbb{E}_{g(x)}[\mathbb{E}[S_{g(x)|x}]]),$$

and obtain

$$\epsilon|T_x| - \epsilon^2 \sqrt{\frac{2M_x|T_x|^3}{N|\mathcal{A}|}}.$$

Summing up over all $x$, and removing the conditioning on the choices of the contexts $x_t$ yields

$$\mathbb{E}[\mathfrak{R}(T)] \geq \epsilon T - \mathbb{E}\left[\sum_{x \in [N]} M_x^{1/2}|T_x|^{3/2}\right]\epsilon^2\sqrt{\frac{2}{N|\mathcal{A}|}}.$$

The above inequality holds for arbitrary $M_x$ so we can consider the following maximization problem

$$\begin{aligned}
\max & \sum_x M_x^{1/2}|T_x|^{3/2} \\
s.t. \quad & \sum_x M_x|T_x| = MT.
\end{aligned}$$

We find the optimal value of $M_x = MT|T_x|/\sum_x |T_x|^2$, and the optimal value of objective is $\sqrt{MT\sum_x |T_x|^2} \leq \sqrt{MT}\sum_x |T_x| = \sqrt{MT^3}$. Therefore, we have

$$\mathbb{E}[\mathfrak{R}(T)] \geq \epsilon T - \epsilon^2\sqrt{\frac{2MT^3}{N|\mathcal{A}|}}.$$

We set $\epsilon = \sqrt{\frac{|\mathcal{A}|N}{4MT}}$ and obtain

$$\mathbb{E}[\mathfrak{R}(T)] \geq 0.14\sqrt{\frac{|\mathcal{A}|NT}{M}} \geq 0.14\sqrt{\frac{|\mathcal{A}|T\ln|\Pi|}{M\ln|\mathcal{A}|}}.$$

Since the expectation is at least $\Omega(\sqrt{\frac{|\mathcal{A}|T\ln|\Pi|}{M\ln|\mathcal{A}|}})$, there must exist an instance which satisfy

$$\mathfrak{R}(T) \geq 0.14\sqrt{\frac{|\mathcal{A}|T\ln|\Pi|}{M\ln|\mathcal{A}|}}.$$

$\square$

## C.5. Proof of Theorem 2.2

*Proof.* By Lemma 2.2 we know that without taking the cost of the queries into account the regret of any algorithm that makes $MT$ observations is lower bounded as:

$$0.14\sqrt{\frac{(|\mathcal{A}| - 1)T\ln(|\Pi|)}{M\ln|\mathcal{A}|}}$$

Adding the cost of observations we have that the regret of any algorithm that makes $MT$ observations is lower bounded by:

$$\mathfrak{R}(T) \geq 0.14\sqrt{\frac{(|\mathcal{A}| - 1)T\ln(|\Pi|)}{M\ln|\mathcal{A}|}} + cMT.$$

If the cost $c > 0$, then we can minimize the right hand side by specifying $M = 0.14^{2/3}(\ln|\mathcal{A}|)^{-\frac{1}{3}}(|\mathcal{A}| - 1)^{\frac{1}{3}}(\ln|\Pi|)^{\frac{1}{3}}c^{-\frac{2}{3}}T^{-\frac{1}{3}}$. If $c = 0$, then we use the bound $M \leq |\mathcal{A}|$ and obtain

$$\mathfrak{R}(T) \geq 0.5\sqrt{T\ln(|\Pi|)/\ln|\mathcal{A}|}.$$

We conclude the proof by combining the two cases:

$$\mathfrak{R}(T) \geq \max\left\{0.14\sqrt{T\ln(|\Pi|)/\ln|\mathcal{A}|}, 0.5(\ln|\mathcal{A}|)^{-\frac{1}{3}}(c|\mathcal{A}|\ln|\Pi|)^{\frac{1}{3}}T^{\frac{2}{3}}\right\}.$$

$\square$

# D. Proofs for Section 3

## D.1. Proof of Theorem 3.1

*Proof.* From Lemma 2.2, we know that the regret of any algorithm with budget $B$ is at least

$$\mathfrak{R}(T) \geq 0.1\sqrt{\frac{|\mathcal{A}|T\ln|\Pi|}{(M + B/T)\ln|\mathcal{A}|}} + cMT.$$

Let $g(M) = \sqrt{\frac{|\mathcal{A}|T\ln|\Pi|}{(M+B/T)\ln|\mathcal{A}|}} + cMT$ and its derivative is

$$g'(M) = -\frac{1}{10}\sqrt{|\mathcal{A}|T\ln(|\Pi|)/\ln|\mathcal{A}|}(M + B/T)^{-\frac{3}{2}} + cT.$$

Its zero point is $M = 10^{-\frac{2}{3}}c^{-\frac{2}{3}}(|\mathcal{A}|\ln|\Pi|)^{\frac{1}{3}}(T\ln|\mathcal{A}|)^{-\frac{1}{3}} - B/T$.

If $B \leq 10^{-\frac{2}{3}}c^{-\frac{2}{3}}(\ln|\mathcal{A}|)^{-\frac{1}{3}}(|\mathcal{A}|\ln|\Pi|)^{\frac{1}{3}}T^{\frac{2}{3}}$, then we have $M \geq 0$. We can plug in the minimizer and obtain

$$\mathfrak{R}(T) \geq 0.3(c|\mathcal{A}|\ln|\Pi|)^{\frac{1}{3}}(\ln|\mathcal{A}|)^{-\frac{1}{3}}T^{\frac{2}{3}} - cB.$$

If $B > 10^{-\frac{2}{3}}c^{-\frac{2}{3}}(\ln|\mathcal{A}|)^{-\frac{1}{3}}(|\mathcal{A}|\ln|\Pi|)^{\frac{1}{3}}T^{\frac{2}{3}}$, then the minimizer will be zero. At this case, we have

$$\mathfrak{R}(T) \geq 0.1T\sqrt{\frac{|\mathcal{A}|\ln|\Pi|}{B\ln|\mathcal{A}|}}.$$

$\square$

## D.2. Proof of Theorem 3.2

Before proving Theorem 3.2, we first present a useful lemma.

**Lemma D.1.** In Algorithm 6, the following inequality holds

$$\sum_{t=1}^{T} \mathbb{E}[|O_t| \mid \mathcal{H}_{t-1}] \leq 1.5c^{-\frac{2}{3}}(|\mathcal{A}|\ln|\Pi|)^{\frac{1}{3}}T^{\frac{2}{3}}.$$

*Proof.* Since each $p_t(a) = \min\left\{1, \sqrt{\eta_t q_t(a)/c_t(a)}\right\}$, we have

$$\sum_{t=1}^{T} \mathbb{E}[|O_t| \mid \mathcal{H}_{t-1}] = \sum_{t=1}^{T}\sum_{a\in\mathcal{A}} p_t(a) \leq \sum_{t=1}^{T}\sum_{a\in\mathcal{A}} \sqrt{\eta_t q_t(a)/c}.$$

Recall that $\eta_t = c^{-\frac{1}{3}}|\mathcal{A}|^{-\frac{1}{3}}\ln^{\frac{2}{3}}(|\Pi|)t^{-\frac{2}{3}}, \forall t \in [T]$. Therefore, we have

$$\sum_{t=1}^{T}\sum_{a\in\mathcal{A}} \sqrt{\eta_t q_t(a)/c} = c^{-\frac{2}{3}}(\ln|\Pi|)^{\frac{1}{3}}|\mathcal{A}|^{-\frac{1}{6}}\sum_{t=1}^{T}\sum_{a\in\mathcal{A}} \sqrt{q_t(a)}t^{-\frac{1}{3}}$$

$$\leq c^{-\frac{2}{3}}(\ln|\Pi|)^{\frac{1}{3}}|\mathcal{A}|^{-\frac{1}{6}}\sum_{t=1}^{T}\sqrt{|\mathcal{A}|\sum_{a\in\mathcal{A}} q_t(a)}t^{-\frac{1}{3}}$$

$$= c^{-\frac{2}{3}}(|\mathcal{A}|\ln|\Pi|)^{\frac{1}{3}}\sum_{t=1}^{T} t^{-\frac{1}{3}}$$

$$\leq 1.5c^{-\frac{2}{3}}(|\mathcal{A}|\ln|\Pi|)^{\frac{1}{3}}T^{\frac{2}{3}}.$$

$\square$

*Proof.* When $B \leq 1.5c^{-\frac{2}{3}}(|\mathcal{A}|\ln|\Pi|)^{\frac{1}{3}}(\ln|\mathcal{A}|)^{-\frac{1}{3}}T^{\frac{2}{3}}$, we run Algorithm 6. Since we have $B$ free observation budgets, we have

$$\mathfrak{R}(T) = \mathbb{E}\left[\sum_{t=1}^{T}(\ell_t(a_t) - \ell_t(\pi^*(x_t))) + c \cdot \max\left\{\sum_{t=1}^{T}|O_t| - B, 0\right\}\right].$$

Denote $Z_t = \sum_{s=1}^{t}|O_s| - \sum_{s=1}^{t}\mathbb{E}[|O_s| \mid \mathcal{H}_{s-1}]$, $Z_0 = 0$. Then $Z_t - Z_{t-1}$ is a martingale difference sequence with $|Z_t - Z_{t-1}| \leq |\mathcal{A}|$. From Azuma-Hoeffding inequality, we have

$$\left|\sum_{t=1}^{T}|O_t| - \sum_{t=1}^{T}\mathbb{E}[|O_t| \mid \mathcal{H}_{t-1}]\right| \leq |\mathcal{A}|\sqrt{2T\ln(2T)},$$

which holds with probability at least $1 - \frac{1}{T}$.

Define the following event

$$\mathcal{E} = \left\{\sum_{t=1}^{T}|O_t| \leq 1.5c^{-\frac{2}{3}}(|\mathcal{A}|\ln|\Pi|)^{\frac{1}{3}}T^{\frac{2}{3}} + |\mathcal{A}|\sqrt{2T\ln(2T)}\right\}.$$

Then from the above analysis, we know that

$$\mathbb{P}(\mathcal{E}) \geq \mathbb{P}\left(\sum_{t=1}^{T}|O_t| \leq \sum_{t=1}^{T}\mathbb{E}[|O_t| \mid \mathcal{H}_{t-1}] + |\mathcal{A}|\sqrt{2T\ln(2T)}\right) \geq 1 - \frac{1}{T},$$

where the first inequality follows from Lemma D.1.

Therefore, we have

$$c \cdot \mathbb{E}\left[\max\{\sum_{t=1}^{T}|O_t| - B, 0\}\right]$$
$$= c \cdot \mathbb{E}\left[\max\{\sum_{t=1}^{T}|O_t| - B, 0\} \mid \mathcal{E}\right]\mathbb{P}(\mathcal{E}) + c \cdot \mathbb{E}\left[\max\{\sum_{t=1}^{T}|O_t| - B, 0\} \mid \overline{\mathcal{E}}\right]\mathbb{P}(\overline{\mathcal{E}})$$
$$\leq c \cdot \mathbb{E}[max\{1.5c^{-\frac{2}{3}}(|\mathcal{A}|\ln|\Pi|)^{\frac{1}{3}}T^{\frac{2}{3}} + |\mathcal{A}|\sqrt{2T\ln(2T)} - B, 0\}] + c|\mathcal{A}|T \times \frac{1}{T}$$
$$\leq 1.5(c|\mathcal{A}|\ln|\Pi|)^{\frac{1}{3}}T^{\frac{2}{3}} - cB + |\mathcal{A}|\sqrt{2T\ln(2T)} + c|\mathcal{A}|,$$

where the last equality follows from the fact that $B \leq 1.5c^{-\frac{2}{3}}(|\mathcal{A}|\ln|\Pi|)^{\frac{1}{3}}T^{\frac{2}{3}}$.

Starting from this, we have:

$$\mathfrak{R}(T) = \mathbb{E}\left[\sum_{t=1}^{T}(\ell_t(a_t) - \ell_t(\pi^*(x_t)))\right] + c \cdot \mathbb{E}\left[\max\{\sum_{t=1}^{T}|O_t| - B, 0\}\right]$$
$$\leq \mathbb{E}\left[\sum_{t=1}^{T}(\ell_t(a_t) - \ell_t(\pi^*(x_t)))\right] + 1.5(c|\mathcal{A}|\ln|\Pi|)^{\frac{1}{3}}T^{\frac{2}{3}} - cB + |\mathcal{A}|\sqrt{2T\ln(2T)} + c|\mathcal{A}|$$
$$\leq 2.5(c|\mathcal{A}|\ln|\Pi|)^{\frac{1}{3}}T^{\frac{2}{3}} - cB + |\mathcal{A}|\sqrt{2T\ln(2T)} + c|\mathcal{A}| + 2.5\sqrt{T\ln|\Pi|} + 2$$

where the last inequality follows from the analysis of Theorem 2.1.

When $B \geq 1.5c^{-\frac{2}{3}}(|\mathcal{A}|\ln|\Pi|)^{\frac{1}{3}}(\ln|\mathcal{A}|)^{-\frac{1}{3}}T^{\frac{2}{3}}$, the meta-controller (Algorithm 2) runs Algorithm 4. From the construction of Algorithm 4, we know that the observation budget $B$ is fully utilized and randomly allocated by Algorithm 5. Due to Proposition E.1, the observation probability for each action $a \in \mathcal{A}$ at each round $t$ is $\frac{B}{|\mathcal{A}|T}$.

From Lemma 2.1, we know that without considering paid observation cost, the learning regret is upper bounded by:

$$\mathfrak{R}(T) = \mathbb{E}\left[\sum_{t=1}^{T}\sum_{a\in\mathcal{A}}(q_t(a) - \mathbb{I}\{\pi^*(x_t) = a\})\ell_t(a)\right] \leq 2\gamma T + \frac{(1-\gamma)\ln|\Pi|}{\eta_{T+1}} + \sum_{t=1}^{T}\eta_t\mathbb{E}\left[\sum_{a\in\mathcal{A}}\frac{q_t(a)}{p_t(a)}\right].$$

Let $p_t(a) = \frac{B}{|\mathcal{A}|T}$ and we obtain

$$\mathfrak{R}(T) \leq 2 + (T+1)\sqrt{\frac{|\mathcal{A}|\ln|\Pi|}{B}}.$$

$\square$

## D.3. Proof of Proposition 4.1

*Proof.* We begin by defining the martingale difference sequence and establishing its key properties.

For each fixed $f \in \mathcal{F}$, define the following sequence

$$M_t(f) = \ell_t(\pi_f(x_t)) - f^*(x_t, \pi_f(x_t))$$

with history $\mathcal{H}_{t-1}$. For any $t \geq 1$, we have

$$\mathbb{E}[M_t(f) \mid \mathcal{H}_{t-1}, x_t] = \mathbb{E}[\ell_t(\pi_f(x_t)) \mid x_t] - f^*(x_t, \pi_f(x_t)) = 0.$$

The first equality holds because given $x_t$, $\pi_f(x_t)$ is deterministic. The second equality follows from the realizability assumption. Therefore, $M_t(f)$ is a martingale difference sequence.

Since $\ell_t(a) \in [0, 1]$ and $f^*(x_t, a) = \mathbb{E}[\ell_t(a)|x_t] \in [0, 1]$, we have:

$$|M_t(f)| = |\ell_t(\pi_f(x_t)) - f^*(x_t, \pi_f(x_t))| \leq 1.$$

Now we establish concentration bounds. For any fixed $f \in \mathcal{F}$ and any $\epsilon > 0$, by Azuma-Hoeffding inequality:

$$\mathbb{P}\left(\left|\sum_{t=1}^{T}M_t(f)\right| \geq \epsilon\right) \leq 2\exp\left(-\frac{\epsilon^2}{2T}\right).$$

Let $X = \sup_{f\in\mathcal{F}}\left|\sum_{t=1}^{T}M_t(f)\right|$. From the union bound, we have for any $\epsilon > 0$,

$$\mathbb{P}(X \geq \epsilon) \leq \sum_{f\in\mathcal{F}}\mathbb{P}\left(\left|\sum_{t=1}^{T}M_t(f)\right| \geq \epsilon\right) \leq 2|\mathcal{F}|\exp\left(-\frac{\epsilon^2}{2T}\right).$$

Choose $\epsilon_0 = \sqrt{2T\log(2|\mathcal{F}|)}$:

$$\mathbb{E}[X] \leq \epsilon_0 + \int_{\epsilon_0}^{\infty}\mathbb{P}(X \geq \epsilon)d\epsilon \leq \epsilon_0 + \int_{\epsilon_0}^{\infty}2|\mathcal{F}|\exp\left(-\frac{\epsilon^2}{2T}\right)d\epsilon.$$

For $\epsilon \geq \epsilon_0$, we have $\frac{\epsilon}{\epsilon_0} \geq 1$, so:

$$\int_{\epsilon_0}^{\infty}\exp\left(-\frac{\epsilon^2}{2T}\right)d\epsilon \leq \int_{\epsilon_0}^{\infty}\frac{\epsilon}{\epsilon_0}\exp\left(-\frac{\epsilon^2}{2T}\right)d\epsilon = \frac{T}{\epsilon_0}\exp\left(-\frac{\epsilon_0^2}{2T}\right).$$

Substitute $\epsilon_0 = \sqrt{2T\log(2|\mathcal{F}|)}$, we obtain

$$\exp\left(-\frac{\epsilon_0^2}{2T}\right) = \exp(-\log(2|\mathcal{F}|)) = \frac{1}{2|\mathcal{F}|}.$$

Therefore:

$$\mathbb{E}[X] \leq \sqrt{2T \log(2|\mathcal{F}|)} + 2|\mathcal{F}| \cdot \frac{T}{\sqrt{2T \log(2|\mathcal{F}|)}} \cdot \frac{1}{2|\mathcal{F}|} = \sqrt{2T \log(2|\mathcal{F}|)} + \frac{\sqrt{T}}{\sqrt{2 \log(2|\mathcal{F}|)}}.$$

Finally, we apply these results to bound $\Delta(T)$. Recall:

$$\Delta(T) = \sum_{t=1}^{T} \ell_t(\pi_{f^*}(x_t)) - \min_{f \in \mathcal{F}} \sum_{t=1}^{T} \ell_t(\pi_f(x_t)).$$

Since $f^* \in \mathcal{F}$, we have:

$$\Delta(T) \leq \sup_{f \in \mathcal{F}} \left| \sum_{t=1}^{T} [\ell_t(\pi_f(x_t)) - f^*(x_t, \pi_f(x_t))] \right| + \sup_{f \in \mathcal{F}} \left| \sum_{t=1}^{T} [\ell_t(\pi_{f^*}(x_t)) - f^*(x_t, \pi_{f^*}(x_t))] \right|$$

$$\leq 2 \sup_{f \in \mathcal{F}} \left| \sum_{t=1}^{T} M_t(f) \right|.$$

Taking expectations and applying the uniform bound theorem:

$$\mathbb{E}[\Delta(T)] \leq 2\mathbb{E} \left[ \sup_{f \in \mathcal{F}} \left| \sum_{t=1}^{T} M_t(f) \right| \right] \leq 2\sqrt{2T \log(2|\mathcal{F}|)}.$$

This completes the proof. □

### D.4. Proof of Lemma 4.1

*Proof.* Note that the loss of the action $a$ can be observed with the probability $p_t(a)$. With a little bit use of notations, we denote the induced distribution of the observation set $O_t$ as $O_t \sim p_t$. We decompose the regret as following:

$$\mathfrak{R}(T)$$
$$= \mathbb{E} \left[ \sum_{t=1}^{T} f^*(x_t, a_t) - f^*(x_t, \pi_{f^*}(x_t)) + \sum_{a \in O_t} c_t(a) \right] + \Delta(T)$$

$$= \mathbb{E} \left[ \sum_{t=1}^{T} f^*(x_t, a_t) - f^*(x_t, \pi_{f^*}(x_t)) - \frac{\eta_t}{4} \left[ \sum_{a' \in O_t} (\hat{f}_t(x_t, a') - f^*(x, a'))^2 + \sum_{a \in O_t} c_t(a) \right] \right] + \Delta(T)$$

$$+ \mathbb{E} \left[ \sum_{t=1}^{T} \frac{\eta_t}{4} \left[ \sum_{a' \in O_t} (\hat{f}_t(x_t, a') - f^*(x, a'))^2 \right] \right] + \Delta(T)$$

$$\leq \sum_{t=1}^{T} \sup_{\substack{a_t^* \in \mathcal{A} \\ f \in \mathcal{F}}} \mathbb{E} \left[ [f(x_t, a_t) - f(x_t, a_t^*)] - \frac{\eta_t}{4} \left[ \sum_{a' \in O_t} (\hat{f}_t(x_t, a') - f(x, a'))^2 + \sum_{a \in O_t} c_t(a) \right] \right] + \Delta(T)$$

$$+ \mathbb{E} \left[ \sum_{t=1}^{T} \frac{\eta_t}{4} \sum_{a' \in O_t} (\hat{f}_t(x_t, a') - f^*(x, a'))^2 \right] + \Delta(T)$$

$$= \sum_{t=1}^{T} Q(q_t, p_t; \hat{f}_t, x_t, \eta_t, \{c_t(a)\}_{a \in \mathcal{A}}) + \mathbb{E} \left[ \sum_{t=1}^{T} \frac{\eta_t}{4} \sum_{a' \in O_t} (\hat{f}_t(x_t, a') - f^*(x, a'))^2 \right] + \Delta(T)$$

$$\leq \sum_{t=1}^{T} Q(q_t, p_t; \hat{f}_t, x_t, \eta_t, \{c_t(a)\}_{a \in \mathcal{A}}) + \frac{\eta_T}{4} \mathbb{E} \left[ \sum_{t=1}^{T} \sum_{a' \in O_t} (\hat{f}_t(x_t, a') - f^*(x, a'))^2 \right] + \Delta(T),$$

where the last inequality is due to the fact that $\eta_t$ is non-decreasing.

Next, since $\mathbb{E}[\ell_t(a)|x_t] = f^*(x_t, a)$ for all $t \in [T]$ and $a \in \mathcal{A}$, we know that

$$\mathbb{E}\left[\sum_{t=1}^{T} \sum_{a' \in O_t} (\hat{f}_t(x_t, a') - f^*(x, a'))^2\right]$$

$$=\mathbb{E}\left[\sum_{t=1}^{T} \sum_{a' \in O_t} (\hat{f}_t(x_t, a') - \ell_t(a'))^2 - \sum_{a' \in O_t} (f^*(x, a') - \ell_t(a'))^2\right]$$

$$\leq \mathbf{Reg}_{Sq}(T),$$

where the last inequality is due to Assumption 4.2.

$\square$

### D.5. Proof of Theorem 4.1

*Proof.* Direct calculation shows that for all $a^* \in \mathcal{A}$

$$\mathbb{E}_{a \sim q_t}\left[f^*(x_t, a) - f^*(x_t, a^*) - \frac{\eta_t}{4}\mathbb{E}_{O_t \sim p_t}\left[\sum_{a' \in O_t} (\hat{f}(x_t, a') - f^*(x_t, a'))^2 + \sum_{a \in O_t} c_t(a)\right]\right]$$

$$= \sum_{a \in \mathcal{A}} q_t(a)f^*(x_t, a) - f^*(x_t, a^*) - \frac{\eta_t}{4}\sum_{a \in \mathcal{A}} p_t(a)(\hat{f}(x_t, a) - f^*(x_t, a))^2 + \sum_{a \in \mathcal{A}} p_t(a)c_t(a).$$

Therefore, taking the gradient over $f^*(x_t, \cdot)$ and we know that

$$\sup_{f^* \in \mathcal{F}}\left[\sum_{a \in \mathcal{A}} q_t(a)f^*(x_t, a) - f^*(x_t, a^*) - \frac{\eta_t}{4}\sum_{a \in \mathcal{A}} p_t(a)(\hat{f}(x_t, a) - f^*(x_t, a))^2 + \sum_{a \in \mathcal{A}} p_t(a)c_t(a)\right]$$

$$= \sum_{a \in \mathcal{A}} q_t(a)\hat{f}(x_t, a) - \hat{f}(x_t, a^*) + \frac{(q_t(a_t^*) - 1)^2}{\eta_t p_t(a_t^*)} + \frac{1}{\eta_t}\sum_{a \neq a_t^*} \frac{(q_t(a))^2}{p_t(a)} + \sum_{a \in \mathcal{A}} p_t(a)c_t(a).$$

Let $\Delta(\mathcal{A})$ be the collection of distributions on $\mathcal{A}$. From Lemma 4.1 we know that we need to choose $q_t$ and $p_t$ to minimize $Q(q_t, p_t; \hat{f}_t, x_t, \eta_t, \{c_t(a)\}_{a \in \mathcal{A}})$. Therefore, we consider the following minimax form:

$$\inf_{\substack{q_t \in \Delta(\mathcal{A}) \\ 0 \leq p_t \leq 1}} Q(q_t, p_t; \hat{f}_t, x_t, \eta_t, \{c_t(a)\}_{a \in \mathcal{A}})$$

$$= \inf_{\substack{q_t \in \Delta(\mathcal{A}) \\ 0 \leq p_t \leq 1}} \sup_{a_t^* \in \mathcal{A}}\left[\sum_{a \in \mathcal{A}} q_t(a)\hat{f}(x_t, a) - \hat{f}(x_t, a_t^*) + \frac{(q_t(a_t^*) - 1)^2}{\eta_t p_t(a_t^*)} + \frac{1}{\eta_t}\sum_{a \neq a_t^*} \frac{(q_t(a))^2}{p_t(a)} + \sum_{a \in \mathcal{A}} p_t(a)c_t(a)\right]$$

$$= \inf_{\substack{q_t \in \Delta(\mathcal{A}) \\ 0 \leq p_t \leq 1}} \sup_{q_t' \in \Delta(\mathcal{A})}\left[\sum_{a \in \mathcal{A}} (q_t(a) - q_t'(a))\hat{f}(x_t, a) + \sum_{a \in \mathcal{A}} p_t(a)c_t(a)\right.$$

$$\left. + \frac{1}{\eta_t}\sum_{a \in \mathcal{A}} \frac{q_t'(a)(q_t(a) - 1)^2}{p_t(a)} + \frac{1}{\eta_t}\sum_{a \in \mathcal{A}} \frac{(q_t(a))^2(1 - q_t'(a))}{p_t(a)}\right]$$

$$= \inf_{0 \leq p_t \leq 1} \inf_{q_t \in \Delta(\mathcal{A})} \sup_{q_t' \in \Delta(\mathcal{A})}\left[\sum_{a \in \mathcal{A}} (q_t(a) - q_t'(a))\hat{f}(x_t, a) + \sum_{a \in \mathcal{A}} p_t(a)c_t(a)\right.$$

$$\left. + \frac{1}{\eta_t}\sum_{a \in \mathcal{A}} \frac{q_t'(a)(q_t(a) - 1)^2}{p_t(a)} + \frac{1}{\eta_t}\sum_{a \in \mathcal{A}} \frac{(q_t(a))^2(1 - q_t'(a))}{p_t(a)}\right]$$

$$= \inf_{0 \leq p_t \leq 1} \sup_{q_t' \in \Delta(\mathcal{A})} \inf_{q_t \in \Delta(\mathcal{A})}\left[\sum_{a \in \mathcal{A}} (q_t(a) - q_t'(a))\hat{f}(x_t, a) + \sum_{a \in \mathcal{A}} p_t(a)c_t(a)\right.$$

$$\left. + \frac{1}{\eta_t}\sum_{a \in \mathcal{A}} \frac{q_t'(a)(q_t(a) - 1)^2}{p_t(a)} + \frac{1}{\eta_t}\sum_{a \in \mathcal{A}} \frac{(q_t(a))^2(1 - q_t'(a))}{p_t(a)}\right],$$

where the last equality is due to Sion's minimax theorem and the fact that the expression is convex in $q_t$.

Let $q_t = q'_t$ and we have

$$
\begin{aligned}
&\inf_{\mathbf{0} \leq p_t \leq \mathbf{1}} \sup_{q'_t \in \Delta(\mathcal{A})} \inf_{q_t \in \Delta(\mathcal{A})} \left[ \sum_{a \in \mathcal{A}} (q_t(a) - q'_t(a))\hat{f}(x_t, a) + \sum_{a \in \mathcal{A}} p_t(a)c_t(a) \right. \\
&\qquad\qquad \left. + \frac{1}{\eta_t} \sum_{a \in \mathcal{A}} \frac{q'_t(a)(q_t(a) - 1)^2}{p_t(a)} + \frac{1}{\eta_t} \sum_{a \in \mathcal{A}} \frac{(q_t(a))^2(1 - q'_t(a))}{p_t(a)} \right] \\
&\leq \inf_{\mathbf{0} \leq p_t \leq \mathbf{1}} \sup_{q'_t \in \Delta(\mathcal{A})} \left[ \frac{1}{\eta_t} \sum_{a \in \mathcal{A}} \frac{q'_t(a)(1 - q'_t(a))}{p_t(a)} + \sum_{a \in \mathcal{A}} p_t(a)c_t(a) \right] \\
&\overset{(a)}{=} \sup_{q'_t \in \Delta(\mathcal{A})} \inf_{\mathbf{0} \leq p_t \leq \mathbf{1}} \left[ \frac{1}{\eta_t} \sum_{a \in \mathcal{A}} \frac{q'_t(a)(1 - q'_t(a))}{p_t(a)} + \sum_{a \in \mathcal{A}} p_t(a)c_t(a) \right] \\
&\overset{(b)}{\leq} \sup_{q'_t \in \Delta(\mathcal{A})} \left[ \sum_{a \in \mathcal{A}} 2\sqrt{q'_t(a)(1 - q'_t(a))c_t(a)/\eta_t} + \frac{1}{\eta_t} + \sum_{a \in \mathcal{A}} c_t(a) \right] \\
&\overset{(c)}{\leq} \sqrt{\frac{2}{\eta_t} \sum_{a \in \mathcal{A}} c_t(a)} + \frac{2}{\eta_t}.
\end{aligned}
$$

Here, the equality (a) is due to Sion's minimax theorem. The inequality (b) attains the infimum at $p_t(a) = \min\left\{1, \sqrt{\frac{q'_t(a)(1 - q'_t(a))}{c_t(a)\eta_t}}\right\}$. The inequality (c) is Cauchy-Schwarz inequality, and if $p_t(a) = 1$ for $a \in \mathcal{A}$, then $\frac{1}{\eta_t} \geq \sum_{a \in \mathcal{A}} c_t(a)$.

Now, in order to guarantee the optimal regret order when $c_t(a) = 0$, we can choose

$$
q'_t(a) = \begin{cases} \frac{1}{|\mathcal{A}| + \gamma(\hat{f}_t(x_t, a) - \hat{f}_t(x_t, a^*_t))}, & a \neq a^*_t, \\ 1 - \sum_{a \neq a^*_t} q_t(a), & a = a^*_t, \end{cases}
$$

which has been proved to be rate-optimal in (Foster & Rakhlin, 2020a).

The above argument shows that

$$
\mathfrak{R}(T) \leq \sum_{t=1}^{T} 2\sqrt{\frac{1}{\eta_t} \sum_{a \in \mathcal{A}} c_t(a)} + \frac{\eta_T}{4} \mathbf{Reg}_{Sq}(T) + \Delta(T).
$$

It remains to show that our selected

$$
\eta_t = \left( \frac{\sum_{s=1}^{t} \sqrt{\sum_{a \in \mathcal{A}} c_s(a)}}{\mathbf{Reg}_{Sq}(T)} \right)^{\frac{2}{3}} + \sqrt{\frac{t}{\mathbf{Reg}_{Sq}(T)}},
$$

can reduce the above expression into claimed form.

We have

$$
\begin{aligned}
\sum_{t=1}^{T} \sqrt{\frac{1}{\eta_t} \sum_{a \in \mathcal{A}} c_t(a)} &\leq (\mathbf{Reg}_{Sq}(T))^{\frac{1}{3}} \sum_{t=1}^{T} \frac{\sqrt{\sum_{a \in \mathcal{A}} c_t(a)}}{\left( \sum_{s=1}^{t} \sqrt{\sum_{a \in \mathcal{A}} c_s(a)} \right)^{\frac{1}{3}}} \\
&\leq \frac{3}{2} (\mathbf{Reg}_{Sq}(T))^{\frac{1}{3}} \left( \sum_{t=1}^{T} \sqrt{\sum_{a \in \mathcal{A}} c_t(a)} \right)^{\frac{2}{3}},
\end{aligned}
$$

where the last inequality is due to Lemma F.1. Though above derivation requires $\sum_{s=1}^{t} \sqrt{\sum_{a \in \mathcal{A}} c_s(a)} > 0$ for all $t \in [T]$, the inequality still holds when $\sum_{s=1}^{t} \sqrt{\sum_{a \in \mathcal{A}} c_s(a)} = 0$ for some $t$.

Then we have

$$\sum_{t=1}^{T} \frac{1}{\eta_t} \leq \sum_{t=1}^{T} \sqrt{\frac{\mathbf{Reg}_{Sq}(T)}{t}}$$

$$\leq 2\sqrt{T\mathbf{Reg}_{Sq}(T)}$$

For the last term, we have

$$\frac{\eta_T}{4}\mathbf{Reg}_{Sq}(T) = (\mathbf{Reg}_{Sq}(T))^{\frac{1}{3}} \left(\sum_{t=1}^{T} \sqrt{\sum_{a \in \mathcal{A}} c_t(a)}\right)^{\frac{2}{3}} + \sqrt{T\mathbf{Reg}_{Sq}(T)}.$$

We conclude the proof by combing all above inequalities. $\qquad\square$

### D.6. Proof of Corollary B.2

*Proof.* From the proof of Theorem 4.1, we only need to bound the term $\sum_{t=1}^{T} Q(q_t, p_t; \hat{f}_t, x_t, \eta_t, \{\bar{c}(a)\}_{a \in \mathcal{A}})$. We have

$$\inf_{\substack{q_t \in \Delta(\mathcal{A}) \\ 0 \leq p_t \leq 1}} Q(q_t, p_t; \hat{f}_t, x_t, \eta_t, \{\bar{c}(a)\}_{a \in \mathcal{A}})$$

$$\leq \sup_{q'_t \in \Delta(\mathcal{A})} \inf_{0 \leq p_t \leq 1} \left[\frac{1}{\eta_t}\sum_{a \in \mathcal{A}} \frac{q'_t(a)(1 - q'_t(a))}{p_t(a)} + \sum_{a \in \mathcal{A}} p_t(a)\bar{c}(a)\right].$$

From the property of sub-gaussian distribution and the union bound, we have

$$\mathbb{P}\left(\bar{c}(a) \leq U_t(a), \forall a \in \mathcal{A}, \forall t \in [T]\right) \geq 1 - \delta/2.$$

We define the event $\mathcal{E} = \{\bar{c}(a) \leq U_t(a), \forall a \in \mathcal{A}, \forall t \in [T]\}$. Therefore, with probability at least $1 - \delta/2$, we have

$$\inf_{\substack{q_t \in \Delta(\mathcal{A}) \\ 0 \leq p_t \leq 1}} Q(q_t, p_t; \hat{f}_t, x_t, \eta_t, \{\bar{c}(a)\}_{a \in \mathcal{A}})$$

$$\leq \sup_{q'_t \in \Delta(\mathcal{A})} \inf_{0 \leq p_t \leq 1} \left[\frac{1}{\eta_t}\sum_{a \in \mathcal{A}} \frac{q'_t(a)(1 - q'_t(a))}{p_t(a)} + \sum_{a \in \mathcal{A}} p_t(a)U_t(a)\right].$$

The observation probability that minimizes the right-hand side is

$$p_t(a) = \min\left\{1, \sqrt{\frac{q_t(a)(1 - q_t(a))}{\eta_t U_t(a)}}\right\}.$$

Given the event $\mathcal{E}$, we have

$$\mathfrak{R}(T) \leq \sum_{t=1}^{T} Q(q_t, p_t; \hat{f}_t, x_t, \eta_t, \{\bar{c}(a)\}_{a \in \mathcal{A}}) + \frac{\eta_T}{4}\mathbf{Reg}_{Sq}(T) + \Delta(T)$$

$$\leq 2(\mathbf{Reg}_{Sq}(T))^{\frac{1}{3}} \left(\sum_{t=1}^{T} \sqrt{\sum_{a \in \mathcal{A}} U_t(a)}\right)^{\frac{2}{3}} + \sqrt{T\mathbf{Reg}_{Sq}(T)}/4 + \Delta(T).$$

from Lemma 4.1. Since

$$U_t(a) \leq \bar{c}(a) + 2\sqrt{\frac{2\log(4|\mathcal{A}|/\delta)}{t}},$$

we have

$$\sum_{t=1}^{T}\sqrt{\sum_{a\in\mathcal{A}}U_t(a)} \leq \sum_{t=1}^{T}\sqrt{\sum_{a\in\mathcal{A}}\bar{c}(a) + 2|\mathcal{A}|\sqrt{\frac{2\log(4|\mathcal{A}|/\delta)}{t}}}$$

$$\leq T\sqrt{\sum_{a\in\mathcal{A}}\bar{c}(a)} + 2|\mathcal{A}|^{\frac{1}{2}}T^{\frac{3}{4}}\left(\log(4|\mathcal{A}|T/\delta)\right)^{\frac{1}{4}}.$$

Combing the above inequalities, we obtain

$$\mathfrak{R}(T) \leq 2(\mathbf{Reg}_{Sq}(T))^{\frac{1}{3}}\left(T\sqrt{\sum_{a\in\mathcal{A}}\bar{c}(a)} + 2|\mathcal{A}|^{\frac{1}{2}}T^{\frac{3}{4}}\left(\log(4|\mathcal{A}|T/\delta)\right)^{\frac{1}{4}}\right)^{\frac{2}{3}} + \sqrt{T\mathbf{Reg}_{Sq}(T)}/4 + \Delta(T)$$

$$\leq 2(\mathbf{Reg}_{Sq}(T))^{\frac{1}{3}}T^{\frac{2}{3}}\left(\sum_{a\in\mathcal{A}}\bar{c}(a)\right)^{\frac{1}{3}}$$

$$+ 3.2(\mathbf{Reg}_{Sq}(T))^{\frac{1}{3}}|\mathcal{A}|^{\frac{1}{3}}T^{\frac{1}{2}}\left(\log(4|\mathcal{A}|T/\delta)\right)^{\frac{1}{6}} + \sqrt{T\mathbf{Reg}_{Sq}(T)}/4 + \Delta(T).$$

The inequality follows from the fact that $(x+y)^{2/3} \leq x^{2/3} + y^{2/3}$ for any $x, y \geq 0$.

$\square$

# E. Algorithms

---

**Algorithm 4** Adversarial Contextual Algorithm with Free Observation Budget I

---

**Input:** Action set $\mathcal{A}$, time horizon $T$, learning rate $\{\eta_t\}_{t=1}^T$, policy class $\Pi$, observation budget $B$, cost $c$, hyperparameter $\gamma$

 1: Call **Algorithm 5**$(\mathcal{A}, T, B)$ to generate observation sets $\{O_t\}_{t=1}^T$ with budget $B$
 2: **for** round $t = 1, 2, \cdots, T$ **do**
 3:     Adversary generates hidden loss set $\{\ell_t(a)\}_{a \in \mathcal{A}}$ and reveals context $x_t$
 4:     Compute the weight $w_t(\pi) = \exp\left(-\eta_t \sum_{s=1}^{t-1} \hat{\ell}_s(\pi(x_s))\right)$
 5:     Compute the sampling probability $q_t(a) = \frac{(1-\gamma) \sum_{\pi \in \Pi : \pi(x_t) = a} w_t(\pi)}{\sum_{\pi \in \Pi} w_t(\pi)} + \frac{\gamma}{|\mathcal{A}|}$
 6:     Sample an action $a_t$ according to $q_t$
 7:     Observe losses $\ell_t(a)$ for $a \in O_t$ using the given budget
 8:     Update $\hat{\ell}_t(a) = \mathbb{I}\{a \in O_t\} \frac{\ell_t(a)}{p_t(a)}, \forall a \in \mathcal{A}$
 9: **end for**

---

**Algorithm 5** Random Subset Selection with Fixed Budget

---

**Input:** action set $\mathcal{A}$, number of rounds $T$, budget $B$ (with $\frac{B}{|\mathcal{A}|T} \in [0, 1]$)

 1: Compute $m = \lfloor B/|\mathcal{A}| \rfloor$ and $r = B - m \times |\mathcal{A}|$
 2: Initialize count array $counts[i] \leftarrow m$ for all $i \in [|\mathcal{A}|]$
 3: Generate random permutation $indices$ of $[|\mathcal{A}|]$
 4: **for** $i = 1$ to $r$ **do**
 5:    $counts[indices[i]] \leftarrow counts[indices[i]] + 1$
 6: **end for**
 7: Initialize $O_t = \emptyset$ for all $t \in [T]$
 8: **for** each element $a \in \mathcal{A}$ with index $i \in [|\mathcal{A}|]$ **do**
 9:    Generate random permutation $round\_indices$ of $[T]$
10:    $selected\_rounds \leftarrow$ first $counts[i]$ elements of $round\_indices$
11:    **for** each round $t \in selected\_rounds$ **do**
12:       Add element $a$ to $O_t$
13:    **end for**
14: **end for**
**Output:** Observation sets $\{O_t\}_{t=1}^T$, where each $O_t \subseteq \mathcal{A}$

---

**Proposition E.1.** The observation sets $\{O_t\}_{t=1}^T$ generated by Algorithm 5 satisfy the following properties:

1. The total number of observations is exactly $B$, i.e., $\sum_{t=1}^T |O_t| = B$.

2. For any round $t \in [T]$ and any action $a \in \mathcal{A}$, the probability that $a$ is observed in round $t$ is

$$\mathbb{P}(a \in O_t) = \frac{B}{|\mathcal{A}|T}.$$

*Proof.* We prove each property separately.

**Proof of Property 1:** In Algorithm 5, each action $a \in \mathcal{A}$ is assigned a count $X_a$ (stored in the array $counts$) representing the total number of times $a$ is to be observed. These counts are determined as follows:

- Compute the base count $m = \lfloor B/|\mathcal{A}| \rfloor$ for each action.

- Compute the remainder $r = B - m|\mathcal{A}|$, representing additional observations to be allocated.

- Randomly select $r$ actions to receive an extra observation, setting $X_a = m + 1$ for these actions and $X_a = m$ for the remaining $|\mathcal{A}| - r$ actions.

---
**Algorithm 6** Adversarial Contextual Algorithm with Free Observation Budget II

---
**Input:** time horizon $T$, action set $\mathcal{A}$, learning rate $\{\eta_t\}_{t=1}^T$, policy space $\Pi$, observation budget $B$, uniform cost $c$, hyperparameter $\gamma$

1: Set the remaining budget $B_1 = B$
2: **for** round $t = 1, 2, \cdots, T$ **do**
3:    Adversary generates hidden loss set $\{\ell_t(a)\}_{a \in \mathcal{A}}$ and reveals context $x_t$
4:    Compute the weight $w_t(\pi) = \exp\left(-\eta_t \sum_{s=1}^{t-1} \hat{\ell}_s(\pi(x_s))\right)$
5:    Compute the sampling probability $q_t(a) = \frac{(1-\gamma)\sum_{\pi \in \Pi: \pi(x_t)=a} w_t(\pi)}{\sum_{\pi \in \Pi} w_t(\pi)} + \frac{\gamma}{|\mathcal{A}|}$
6:    Sample an action $a_t$ according to $q_t$
7:    For each $a$, put $a$ into the observation set $O_t$ with probability $p_t(a) = \min\left\{1, \sqrt{\frac{\eta_t q_t(a)}{c}}\right\}$
8:    Observe losses $\ell_t(a)$ for $a \in O_t$ using budget $B_t$
9:    Pay cost $(|O_t| - B_t)c$ for excess observations if needed
10:   Update the remaining budget $B_{t+1} = \max\{B_t - |O_t|, 0\}$
11:   Update $\hat{\ell}_t(a) = \mathbb{I}\{a \in O_t\}\frac{\ell_t(a)}{p_t(a)}, \forall a \in \mathcal{A}$
12: **end for**

---

---
**Algorithm 7** Contextual Bandit Algorithm with Unknown Costs

---
**Input:** time horizon $T$, action set $\mathcal{A}$, a regression oracle $\mathbf{Alg}_{Sq}$, hyperparameter $\gamma$

1: **for** epoch $t = 1, 2, \cdots, T$ **do**
2:    A loss vector $\{\ell_t(a)\}_{a \in \mathcal{A}}$ is generated by the adversary but not disclosed to the learner
3:    A context $x_t$ is generated by the adversary and revealed to the learner
4:    Obtain an estimator $\hat{f}_t$ from the online oracle $\mathbf{Alg}_{Sq}$
5:    Compute the sampling probability

$$q_t(a) = \begin{cases} \frac{1}{|\mathcal{A}|+\gamma(\hat{f}_t(x_t,a)-\hat{f}_t(x_t,a_t^*))}, & a \neq a_t^*, \\ 1 - \sum_{a \neq a_t^*} q_t(a), & a = a_t^*, \end{cases}$$

   where $a_t^* = \arg\min_{a \in \mathcal{A}} \hat{f}_t(x_t, a)$.
6:    Sample an action $a_t \sim q_t(a)$ and take the action
7:    Compute

$$\eta_t = \left(\frac{\sum_{s=1}^t \sqrt{\sum_{a \in \mathcal{A}} U_s(a)}}{\mathbf{Reg}_{Sq}(T)}\right)^{\frac{2}{3}} + \sqrt{\frac{t}{\mathbf{Reg}_{Sq}(T)}},$$

8:    Compute $p_t(a) = \min\left\{1, \sqrt{\frac{q_t(a)(1-q_t(a))}{\eta_t U_t(a)}}\right\}$
9:    For each $a$, with probability $p_t(a)$, put the action $a$ into the observation set $O_t$
10:   Pay the cost $\sum_{a \in O_t} c_t(a)$ to observe a loss $\ell_t(a), \forall a \in O_t$
11:   Obtain the stochastic observation costs $c_t(a)$ for all $a \in \mathcal{A}$
12:   Update $\mathbf{Alg}_{Sq}$ with tuples $\{(x_t, a, \ell_t(a))\}_{a \in O_t}$
13:   Update UCB $U_t(a)$ for all $a \in \mathcal{A}$ according to (2)
14: **end for**

---

---

**Algorithm 8** Adversarial Contextual Algorithm with Limited Budgets

---

**Input:** action set $\mathcal{A}$, learning rate $\eta_t$, policy space $\Pi$

1: **for** epoch $t = 1, 2, \cdots$ **do**
2:     A loss vector $\{\ell_t(a)\}_{a \in \mathcal{A}}$ is generated by the adversary but not disclosed to the learner
3:     A context $x_t$ is generated by the adversary and revealed to the learner
4:     Obtain the observation budget $M_t$
5:     Compute the weight

$$w_t(\pi) = \exp\left(-\eta_t \sum_{s=1}^{t-1} \hat{\ell}_s(\pi(x_s))\right)$$

6:     Compute the sampling probability $q_t(a) = \frac{\sum_{\pi \in \Pi : \pi(x_t) = a} w_t(\pi)}{\sum_{\pi \in \Pi} w_t(\pi)}$
7:     Sample an action $a_t$ according to $q_t$
8:     Uniformly sample $M_t$ actions on $|\mathcal{A}|$ without replacement
9:     Denote the observation set as $O_t$ and update

$$\forall a \in \mathcal{A}, \hat{\ell}_t(a) = \mathbb{I}\{a \in O_t\} \frac{\ell_t(a)}{p_t(a)},$$

     where $p_t(a) = \frac{M_t}{|\mathcal{A}|}$
10: **end for**

---

By construction, the sum of all counts is:

$$\sum_{a \in \mathcal{A}} X_a = (|\mathcal{A}| - r) \cdot m + r \cdot (m + 1) = m|\mathcal{A}| + r = B.$$

Each action $a$ is then assigned to exactly $X_a$ distinct rounds through random permutation. Therefore, the total number of observations across all rounds is:

$$\sum_{t=1}^{T} |O_t| = \sum_{a \in \mathcal{A}} X_a = B.$$

This establishes Property 1.

**Proof of Property 2:** For any fixed action $a \in \mathcal{A}$ and round $t \in [T]$, the count $X_a$ is a random variable with distribution:

$$\mathbb{P}(X_a = m) = 1 - \frac{r}{|\mathcal{A}|}, \quad \mathbb{P}(X_a = m + 1) = \frac{r}{|\mathcal{A}|}.$$

Given $X_a$, the probability that $a$ is assigned to a specific round $t$ is $X_a/T$, as rounds are selected uniformly at random without replacement. Thus, by the law of total expectation:

$$\mathbb{P}(a \in O_t) = \mathbb{E}\left[\frac{X_a}{T}\right] = \frac{\mathbb{E}[X_a]}{T}.$$

The expected count is:

$$\mathbb{E}[X_a] = m \cdot \left(1 - \frac{r}{|\mathcal{A}|}\right) + (m + 1) \cdot \frac{r}{|\mathcal{A}|} = m + \frac{r}{|\mathcal{A}|}.$$

Since $m|\mathcal{A}| + r = B$, we have:

$$m + \frac{r}{|\mathcal{A}|} = \frac{B}{|\mathcal{A}|}.$$

Therefore,

$$\mathbb{E}[X_a] = \frac{B}{|\mathcal{A}|}, \quad \text{and} \quad \mathbb{P}(a \in O_t) = \frac{B}{|\mathcal{A}|T}.$$

This establishes Property 2 and completes the proof. □

# F. Additional Materials

**Lemma F.1** (Lemma 4.13 in (Orabona, 2019)). Let $s_0 \geq 0$ and $f : (0, +\infty) \to (0, +\infty)$ a non-increasing function. Then

$$\sum_{t=1}^{T} s_t f(s_0 + \sum_{i=1}^{t} s_i) \leq \int_{s_0}^{\sum_{t=1}^{T} s_t} f(x)dx.$$

**Lemma F.2** (Azuma's Inequality). Let $\{X_\tau\}_{\tau=1}^{n}$ be a martingale difference sequence with respect to a filtration $\{\mathcal{F}_\tau\}_{\tau=0}^{n}$, i.e., $\mathbb{E}[X_\tau \mid \mathcal{F}_{\tau-1}] = 0$ for all $\tau$. Assume $a_\tau \leq X_\tau \leq b_\tau$ a.s. for $\tau = 1, \ldots, n$. Then for any $\iota \in (0, 1)$,

$$\mathbb{P}\left( \left| \frac{1}{n} \sum_{\tau=1}^{n} X_\tau \right| \geq \frac{1}{n} \sqrt{\frac{1}{2} \ln(2/\iota) \sum_{\tau=1}^{n} (b_\tau - a_\tau)^2} \right) \leq \iota.$$

