# OpenReview forum: "The Cost of Information: Phase Transitions in Contextual Bandits with Paid Observations"
_ICML.cc/2026/Conference — ICML 2026 regular_

### Official Review · Reviewer_Up5K · 2026-02-24

**Soundness:** 4
**Presentation:** 4
**Significance:** 3
**Originality:** 3
**Overall Recommendation:** 5
**Confidence:** 4

**Summary:**

This paper studies an extension of the multi-armed bandits with paid observation problem to the contextual bandit setting. In this setting, in addition to choosing an arm to pull, the learner can choose a set of arms of which the potential loss if they had been chosen is disclosed (this set may or may not contain the actually chosen action) but each arm has a cost. The goal is to minimize the sum of losses and costs. The authors propose a minimax optimal learning procedure in the adversarial setting and a more computationally efficient algorithm leveraging an online regression oracle in the stochastic setting. Additionally, they provide detailed analysis of the regret lower bound of the problem when introducing varying quantities of free observation budget and propose an algorithm able to detect the current regime and use the method best adapted to the current case, thus achieving near-optimal performance no matter the current budget regime. Finally, the authors perform numerical experiments to display the impact of the different parameters on the regret empirically.

**Compliance With Llm Reviewing Policy:**

Affirmed.

**Final Justification:**

My main concern was about the experimental study and has been properly addressed with additional experiments. The authors clarified their contribution and pointed toward interesting extensions I asked about that were already in their appendix.

**Key Questions For Authors:**

1) The numerical study focuses on the regret of proposed algorithms. Could you provide a comparison with a baseline (e.g. standard contextual bandit algorithm unaware of the cost of observation) and with other algorithms from the literature.
2) The paper assumes a fixed, known observation cost c in its optimality results. Could the authors comment on happens if we relax this strong assumption on the cost by introducing non-uniform constant cost or stochastic cost?

**Limitations:**

No discussed potential negative societal impact but irrelevant for this work. The limitation are generally discussed throughout but should be made more apparent by including them (or repeating them) at the end of the paper.

**Strengths And Weaknesses:**

Strengths:
- All the results are properly justified and the proof in appendix seem sound. The authors study the problem and provide results with the most general assumptions that can be expected in this setting.
- In addition to the strong results in the general setting, the authors provide a computationally efficient algorithm that make their method more usable in practice.
- The paper is well written and clear overall. It may be useful to revise the introduction of the setting that is a bit confusing in the introduction but clarified in section 2. Also make sure to format line 6 of algorithm 3 which is very hard to read.
- The paper provides a proper analysis of a general contextual bandit setting that was not studied before (although previous work on linear bandit with costly reward observation exists)

Weaknesses:
- The numerical study is rather weak and lack comparison with other algorithms of the literature (for instance Tucker et al. 2022) in particular when you are focusing on linear bandits in the experiments.
- Not really a weakness, but in terms of novel technical contribution the paper is a bit thin. However, I think they do a good job on the conceptual and problem formulation side which I have not seen before in this particular setting.

---

> ### Author Rebuttal · Authors · 2026-03-30
>
> We thank the reviewer for the constructive feedback. We address each point below.
>
> **Numerical study lacks comparison with other algorithms.**
> We have added a comparison with linUCB and Tucker et al. in Section B. The detailed setup and discussion are provided in our response to Reviewer 1gCG. The key result is:
>
> | Algorithm | Final Average Cumulative Regret |
> |-----------|----------------------------------|
> | linUCB    | $5003.32 \pm 1.04$ |
> | Tucker    | $3697.13 \pm 26.80$ |
> | Algorithm 3      | $2899.75 \pm 48.80$ |
>
> Algorithm 3 achieves the lowest regret by jointly optimizing the exploration-exploitation and cost-information trade-offs.
>
> **Technical contribution is somewhat thin.**
> We appreciate your acknowledgment of the conceptual and problem-formulation contributions. As detailed in our response to Reviewer gBcu, the paper offers several non-trivial technical contributions beyond the formulation:
>
> - **Lower bound construction:** A novel reduction from contextual bandits to multiple independent MAB instances shows that observation costs raise the minimax regret from $\Theta(T^{1/2})$ to $\Theta(T^{2/3})$.
> - **Free-budget analysis:** An entirely new setting with a critical budget threshold, a meta-controller that adapts between two strategies, and matching upper and lower bounds revealing a phase transition.
> - **Oracle-efficient generalization:** An algorithm handling general function spaces via an online regression oracle, with technical innovations including convex-concave reformulations, Sion's minimax theorem, and closed-form observation probabilities (see Page 7, lines 361-370).
> - **Extensions to uninformed costs and time-varying budgets** (Appendix C.2 and Corollary C.1), demonstrating the flexibility of our framework.
>
> **Relaxing the fixed, known observation cost assumption.**
> The uniform known cost $c$ is used for clarity in the free-budget analysis. We also consider extensions:
>
> * **Non-uniform costs:**
> Our analysis in Theorems 2.1 and 4.1 naturally accommodates time-varying and action-dependent costs $c_t(a)$, as the bounds are expressed in terms of $\sum_{t=1}^T \sqrt{\sum_{a\in\mathcal{A}} c_t(a)}$.
>
> * **Stochastic unknown costs:**
> Appendix C.2 (Corollary C.2) explicitly handles i.i.d. costs drawn from a sub-Gaussian distribution with unknown mean. Using UCB-based cost estimates, we show that the additional regret due to cost uncertainty is only $\tilde{\mathcal{O}}(\sqrt{T})$, which is dominated by the $\tilde{\mathcal{O}}(T^{2/3})$ main term.
>
>
>
> **Potential negative societal impact and discussion of limitations.**
> As you point out, discussion of potential negative societal impact is irrelevant for this work. To make limitations more apparent, we revise the Conclusion as follows:
>
> "
> Several future directions remain open. First, extending the analysis to heterogeneous observation costs in the free budget setting is an important step: while our lower bound and meta-controller currently assume a uniform cost $c$ across actions and rounds, many practical scenarios involve non-uniform or time-varying costs, and handling such general cost structures would require a more nuanced exploration-cost trade-off. Extensions to non-stationary or combinatorial action spaces could broaden applicability. Practically, integrating other oracle-efficient approach in fully adversarial settings and conducting large-scale experiments are promising. Finally, links to multi-agent systems and differential privacy offer rich avenues for further work.
> "

---

> > ### Author Rebuttal · Reviewer_Up5K · 2026-04-01
> >
> > Thank you for your answer. My main concern was about the experimental study and has been properly addressed with additional experiments. The authors clarified their contribution and pointed toward interesting extensions I asked about that were already in their appendix. I suggest that the authors refer to this appendix section in the main text (not the case currently).
> > I will increase my score.

---

> > > ### Author Response · Authors · 2026-04-04
> > >
> > > Thank you very much for your positive feedback and for raising your score. We will follow your suggestion and add a reference to the relevant appendix section in the main text.

---

### Official Review · Reviewer_1gCG · 2026-03-04

**Soundness:** 3
**Presentation:** 3
**Significance:** 3
**Originality:** 3
**Overall Recommendation:** 5
**Confidence:** 4

**Summary:**

The paper studies contextual bandits with paid observations, where the learner can actively acquire additional feedback at a cost. The paper appears to analyze a major challenge in such settings: balancing the benefit of acquiring additional information with the cost incurred by observing it.

At each round, a context is revealed adversarially together with a cost associated with observing the loss of each action. The learner may then choose a subset of actions for which it pays to observe the corresponding losses, while selecting an action according to a policy from a predefined policy class. The learner’s objective is to minimize the cumulative regret, which accounts for both the discrepancy between the learner’s actions and the optimal policy in hindsight, and the cost incurred from acquiring additional observations.

The authors first propose Algorithm 1, which can be viewed as an adaptation of EXP3-style policy-weighting methods to the paid-observation contextual bandit setting. The algorithm incorporates observation costs into the sampling and estimation procedure and achieves a regret bound that matches the established lower bound for this setting.

The paper further investigates a phase transition phenomenon when the learner is allowed a free observation budget. The regret behavior depends critically on the amount of available budget. To handle this regime, the authors introduce Algorithm 2, a meta-controller that selects strategies depending on the available observation budget and achieves regret bounds matching the corresponding lower bounds.

The paper also studies a stochastic setting with paid observations under a realizability assumption. In this case, the authors propose Algorithm 3, an oracle-efficient method based on function approximation and regression oracles. This algorithm adapts the ideas of Algorithm 1 to the stochastic setting while maintaining computational efficiency, and the authors provide regret guarantees for this framework.

Finally, the paper includes numerical experiments evaluating Algorithms 1 and 3 on synthetic datasets, illustrating the regret behavior under different parameter configurations and supporting the theoretical findings.

**Compliance With Llm Reviewing Policy:**

Affirmed.

**Final Justification:**

This is a good paper from several perspectives: it addresses an interesting problem, provides clear contributions, and is well written. I support its acceptance.

**Key Questions For Authors:**

Please feel free to address any of the points mentioned above.

**Limitations:**

yes

**Strengths And Weaknesses:**

**Strengths:**

The paper is clearly written and overall enjoyable to read. The exposition is well structured and the presentation of the problem, algorithms, and theoretical results is easy to follow.

The problem setting is well motivated, with several concrete real-world examples illustrating the relevance of contextual bandits with costly observations. In particular, the phase transition phenomenon arising when free observation budgets are introduced is an interesting conceptual contribution that helps clarify the trade-offs between observation cost and learning performance.

The proposed algorithms are carefully described and supported by rigorous theoretical analysis. The paper provides both regret upper bounds and matching lower bounds, which strengthens the theoretical contribution.

Finally, the problem is studied under several complementary perspectives, including adversarial and stochastic settings, as well as extensions incorporating free observation budgets. This broad treatment provides a comprehensive view of the problem and its different regimes.

**Weaknesses:**

I do not see major weaknesses in the paper; the following points are mainly remarks and suggestions for improvement.

**Computational scalability in the adversarial setting.** The adversarial algorithm relies on maintaining weights over the entire policy class. As a consequence, Algorithm~1 requires enumerating the policy space, which may become computationally infeasible when the policy class is large. Although the paper later proposes an oracle-efficient method in the stochastic setting, this limitation remains in the adversarial framework and may restrict the practical applicability of the approach. It would be interesting to discuss whether similar oracle-based techniques could be incorporated in the adversarial setting.

**Limited experimental evaluation.** The experimental section is relatively limited. The experiments are conducted only on synthetic data and mainly illustrate the expected sublinear regret behavior. While these results are consistent with the theoretical analysis, additional experiments on real-world datasets or comparisons with relevant baselines would further strengthen the empirical validation of the proposed methods.

**Lack of proof intuition in the main text.**  Theoretical results are clearly stated, but the paper could benefit from including short proof sketches or intuitive explanations of the main arguments in the main text. This would help readers better understand the key ideas behind the analysis.

**Minor error.** In the filtration definition on page 3 (right column, around line 141), it seems that the loss values should only be revealed for actions in the observation set $O_t$ (i.e., $\{\ell_t(a)\}_{a \in O_t}$), rather than for all actions.

---

> ### Author Rebuttal · Authors · 2026-03-30
>
> We thank the reviewer for the thoughtful comments and constructive suggestions. We address each point below.
>
> **Computational scalability in the adversarial setting.**
> The reviewer correctly notes that Algorithm 1 requires maintaining weights over $\Pi$, which can be computationally prohibitive when $\Pi$ is large. This limitation is inherent to the adversarial setting, where no structural assumptions on the policy space are made. Since our adversarial algorithm follows the exponential-weight framework, efficiency-improving techniques developed for EXP-type algorithms can be naturally incorporated.
>
> We have addressed this challenge by introducing an oracle-efficient algorithm (Algorithm 3) that operates under a realizability assumption and achieves computational efficiency even for infinite function classes. Extending oracle-based methods to the fully adversarial setting (without realizability) is a nontrivial open problem, as the lack of structural assumptions makes it difficult to avoid explicit enumeration. We agree this is an interesting direction and have added a remark in the conclusion.
>
> **Limited experimental evaluation.**
> We add a new experiment to compare our method with baselines and present the following text in Section B:
>
> "
> We compare our algorithms with several baselines: linUCB, a standard contextual bandit algorithm that does not account for observation costs, and Tucker et al., the only existing work on linear contextual bandits with costly observations. The contextual feature dimension is $d = 10$, the number of arms is $|\mathcal{A}| = 5$, and the horizon is $T = 10{,}000$. Since Tucker et al. consider fixed costs, we set $c = 0.5$ for all algorithms. All experiments are repeated for $50$ independent runs.
>
> Table: Final Average Cumulative Regret of Different Algorithms
>
> | Algorithm | Final Average Cumulative Regret |
> |-----------|----------------------------------|
> | linUCB    | $5003.32 \pm 1.04$ |
> | Tucker    | $3697.13 \pm 26.80$ |
> | Algorithm 3      | $2899.75 \pm 48.80$ |
>
> linUCB incurs the highest cost-inclusive regret because it does not account for observation costs in its decision-making, resulting in excessive spending on observations; its low variance reflects this deterministic observation strategy. Tucker's algorithm achieves lower regret by incorporating cost awareness, but is outperformed by Algorithm 3. Our method obtains the lowest regret by jointly optimizing the exploration-exploitation and cost-information trade-offs. The slightly higher variance of Algorithm 3 reflects its adaptive observation strategy, which introduces more variability but achieves better mean performance.
> "
>
>
>
> **Lack of proof intuition in the main text.**
> We agree that adding short proof sketches improves accessibility. Our main text already contains several intuitive discussions. For example, after Theorem 2.1, we derive the optimal observation probability $p_t(a)=\min(1,\sqrt{\eta_t q_t(a)/c_t(a)} )$ in closed form and explain how the cost term shifts the regret from $\tilde\Theta(T^{1/2})$ to $\tilde\Theta(T^{2/3})$. Similarly, Lemma 2.2 provides an intuitive argument: with $M$ observations per round on average, the regret must scale as $\Omega\left(\sqrt{|\mathcal{A}|T\ln|\Pi|/(M\ln|\mathcal{A}|)}\right)$; combining this with the cost yields the final lower bound. We have added further proof sketches for Theorems 3.1 and 4.1 in the revision.
>
> Specifically, the main text provides proof intuition for:
>
> * **Lemma 2.1**: how it guides the design of observation probabilities to balance cost and performance.
> * **Theorem 2.1**: closed-form optimal observation probability and the regret shift.
> * **Theorem 2.2**: via Lemma 2.2's reduction argument.
> * **Theorem 3.1**: detailed description of the phase transition mechanism.
> * **Algorithm 2**: rationale for the meta-controller's strategy selection.
> * **Theorem 4.1**: via Lemma 4.1's minimax framework for choosing $q_t$ and $p_t$.
> * **Corollary C.2**: how UCBs handle unknown costs.
>
>
>
> **Minor error in the filtration definition.**
> We thank the reviewer for catching this. The filtration should be defined with $(\ell_s(a))_{a\in O_s}$.

---

> > ### Author Rebuttal · Reviewer_1gCG · 2026-04-02
> >
> > I thank the authors for their response and am satisfied.

---

> > > ### Author Response · Authors · 2026-04-04
> > >
> > > Thank you for your initial support and for confirming that you are satisfied with our response. We greatly appreciate your positive assessment.

---

### Official Review · Reviewer_CA4a · 2026-03-05

**Soundness:** 3
**Presentation:** 2
**Significance:** 3
**Originality:** 3
**Overall Recommendation:** 5
**Confidence:** 4

**Summary:**

In this work, the authors consider the problem of adversarial contextual bandits with paid observations. This means that at each round, the learner doesn't necessarily observe the loss associated with the action they played, but rather, they can decide on a subset of the arms to sample, which comes at a cost that adds to the regret. Then, the authors consider a variant of the problem where the learner has access to a fixed amount of free observations, and they show that there is a phase switch in the regret rate depending on the cost of the observations and the number of free observations given to the learner. Interestingly, a very large number of bandit frameworks are represented in this formulation of the problem, which means that the paper provides results that are applicable to a variety of problems (which might have only been studied in the standard MAB framework and not the contextual one).

The authors propose algorithms and proofs for several settings, notably the regime where the learner has to pay a fixed cost $c$ for each observation and the regime where the learner has access to a limited free exploration budget.

They also provide some experiments.

**Compliance With Llm Reviewing Policy:**

Affirmed.

**Final Justification:**

The authors have addressed my few concerns in the rebuttal. I am happy to recommend acceptance of this paper and have adjusted my score.

**Key Questions For Authors:**

Could you discuss (and add to the paper) why it is important use extra exploration in the algorithm using the $\gamma$ parameter?

Could you better explain what $\Delta(T)$ (page 6) represents? Is it common to phrase the problem this way rather than defining the regret against a class of realizable policies? What is the benefit of using this representation?

You discuss the effect of non-constant costs in the appendix. Do you think that your method would generalize to different costs (maybe fixed but arm-dependent, arbitrary costs but with some monotonicity assumption)?




I would also point out that the proposed framework of the bandits with free observations is so general that it includes other frameworks as special cases: if the number of observations is $B = T$ and the learner is limited to one observation per round, then this framework represents the Decoupled Exploration and Exploitation in MAB problem (Decoupling Exploration and Exploitation in Multi-Armed Bandits, Avner et al 2012). With $B = 2T$ and two observations per round, it becomes possible to adapt to a small effective range of the losses (Adaptation to Easy Data in Prediction with Limited Advice, Thune and Seldin 2018).
While this is not the core of the paper, it might be of interest to the authors to check what happens in these settings as well.


======
Minor typos and feedback:

On page 3 (around l143, right column), the sentence about whether $\ell_t(a_t)$ is observed isn't super clear. Saying " $\ell_t(a_t)$ may not be observed" is ambiguous, as it could signify that the learner isn't allowed to pick it (seeing "may" as a permission). Consider changing this into  $\ell_t(a_t)$ might not be observed [...], and thus its loss would not be retrieved".

On page 5 (top of the right column), what you write as the choice of $B$ doesn't quite match what you write as the cases in Theorem 3.1. Is this normal?

On page 6 (top of the right column), consider using punctuation at the end of your equation.

In the figures on page 8, consider adding the correct punctuation in the captions.

In the Appendix, on page 16, the page alignment isn't so great. Consider moving things around to make the page look better (maybe writing the first statement in the last eqarray on page 16 and the first on page 17, as the first equation being on the left rather than above the second one is enough to fix this display issue?)

**Limitations:**

yes

**Strengths And Weaknesses:**

This work proposes a nice generalized interface to frame wide variety of multi-armed bandits problems. The proposed algorithms are based on exponential weights algorithms, which is a very standard approach and what is used for most of the non-contextual versions of these problems: The article provides many results which align well the rest of the literature, and the proofs are adequately detailed in the appendix.
The presentation of the extra motivating examples in the appendix is very useful and makes a good case for the usefulness of these results.

One part that would perhaps benefit from being more detailed is the presentation of the problem setting: as it stands, it isn't very clear why the contextual bandits problem is more challenging than the standard MAB problem. It would be very useful to add a more detailed discussion (or maybe a reference to some seminal work on contextual bandits) so that the reader gets a sense of why the adaptation to context isn't trivial and is worth being studied. This also means that the definition of $\Delta(T)$ on page 6 isn't very clear and that the concept of realizability should be better explained.

As it stands, the paper proposes interesting results that are mathematically sound, but the presentation of the paper makes it difficult for people that aren't very knowledgeable in the specific challenges of contextual bandits to understand.

---

> ### Author Rebuttal · Authors · 2026-03-30
>
> Thank you for your thoughtful review. We address your comments below.
>
> **Why the contextual setting is more challenging.**
> The main difficulty lies in the *policy space* $\Pi$: unlike the finite-arm MAB case, $\Pi$ can be arbitrarily large (even infinite). Standard exponential-weight methods that maintain a weight per action become infeasible when policies are not simply actions. In the function-approximation setting (Section 4), we consider general function classes where even linear functions already pose significant challenges. Tucker et al. (2022) study linear contextual bandits with paid observations but provide no theoretical guarantees. In contrast, our oracle-efficient algorithm (Algorithm 3) handles arbitrary function spaces, leverages an online regression oracle, and delivers rigorous regret bounds. The technical innovations (integrating the decision-estimation coefficient with cost-sensitive observations and deriving a novel closed-form observation probability) are detailed on page 7, lines 361-370.
>
> **Role of the exploration parameter $\gamma$.**
> In Algorithm 1, $\gamma$ ensures that the sampling probability $q_t(a)$ is bounded away from zero ($q_t(a) \ge \gamma/|\mathcal{A}|$). This uniform exploration is essential for controlling the variance of the importance-weighted estimators, preventing them from becoming arbitrarily large. It is standard in EXP4-style algorithms; we set $\gamma = 1/T$ in the final bound to balance exploration and regret.
>
> **Meaning of $\Delta(T)$ on page 6.**
> The term $\Delta(T)$ bridges two different benchmarks. The optimal policy $\pi*$ is defined in hindsight as $\arg\min_{\pi\in\Pi}\sum_{t=1}^T \ell_t(\pi(x_t))$, which can exploit the randomness of realized losses. In contrast, $\pi_{f*}$ is optimal in the sense of conditional expectation, since the function class $\mathcal{F}$ models $\mathbb{E}[\ell_t(a)|x_t]$. We adopt $\pi*$ as the benchmark to align with the regret definition in Eq. (1). The term $\Delta(T) = \sum_{t=1}^T [\ell_t(\pi_{f*}(x_t)) - \ell_t(\pi^*(x_t))]$ quantifies the gap between these two benchmarks. Proposition 4.1 shows $\Delta(T) \le O(\sqrt{T \log |\mathcal{F}|})$, which is dominated by the $\tilde{O}(T^{2/3})$ main regret term and thus does not affect the regret order.
>
>
>
> **Extension to non-constant costs.**
> Our algorithms (Algorithm 1 and Algorithm 3) handle arbitrary time- and action-dependent costs $\{c_t(a)\}$ without any monotonicity or stationarity assumptions. The theoretical guarantees in Theorems 2.1 and 4.1 apply directly to general cost vectors. Fixed arm-dependent costs are a special case, and we also address the uninformed costs setting (where costs are unknown) in Appendix C. We hope this clarifies the generality of our approach and are happy to discuss further.
>
>
>
> **Connections to related works.**
> Thanks for your suggestions. We add the following comparisons in Appendix C.
>
> *Comparison to Avner et al. (2012).* Their decoupled exploration setting fixes the amount of information to one reward observation per round. In contrast, our framework allows the learner to select an arbitrary subset of arms to observe at a cost that may vary across arms and over time. This additional flexibility introduces the fundamental cost-information trade-off that drives the phase transition in regret rate. Moreover, our analysis handles adversarial contexts and general policy classes, whereas Avner et al. (2012) focus on the non-contextual setting.
>
> *Comparison to Thune & Seldin (2018).* Their setting provides exactly one free extra observation per round in a non-contextual prediction problem; SODA achieves regret $O(\epsilon\sqrt{|\mathcal{A}|T \ln |\mathcal{A}|})$ against adversarial losses with range $\epsilon$. While their setting resembles ours when $B = 2T$, the results are not directly comparable due to different assumptions on observation structure and loss range.
>
>
> **Minor typos and formatting.**
> Thanks for your constructive feedback. We follow your suggestions and correct typos.

---

> > ### Author Rebuttal · Reviewer_CA4a · 2026-04-04
> >
> > Thank you for your answers. I don't have any further comments at this point.

---

> > > ### Author Response · Authors · 2026-04-04
> > >
> > > Thank you for the positive acknowledgment and for your helpful feedback. We are glad that you have no further comments.

---

### Official Review · Reviewer_gBcu · 2026-03-10

**Soundness:** 3
**Presentation:** 3
**Significance:** 2
**Originality:** 3
**Overall Recommendation:** 4
**Confidence:** 3

**Summary:**

The authors extend the setting of Seldin et al. (2014) to the contextual case with a non-oblivious adversary. They propose a bandit algorithm that extends the EXP3-style method of Seldin et al. (2014) to a contextual EXP4-style algorithm. For the finite-arm setting, they establish both lower and upper regret bounds.

They further consider a setting in which the agent has an additional free-observation budget $B$, while observation costs are constant over time and across actions. They show that when the budget is small relative to the cost $c$ and the horizon $T$, the regret bounds remain unchanged, whereas for sufficiently large budgets, asymptotic improvements become possible.

In the final part of the paper, the results are extended to a setting with structural assumptions on the policy class.

**Compliance With Llm Reviewing Policy:**

Affirmed.

**Final Justification:**

I have nothing to add beyond what I wrote in the acknowledgment

**Key Questions For Authors:**

Related to the weaknesses mentioned above, I encourage the authors to explain more precisely in what way their extension requires novel ideas beyond simply combining an EXP4-type analysis with the framework of Seldin et al. (2014). In particular, the paper would benefit from a clearer discussion of which parts of the argument are genuinely new and which ones are adaptations of existing techniques.

I have a similar request regarding the free-observation-budget setting. The analysis appears to rely on the assumption that the observation cost is constant over time and across actions. It would be helpful if the authors could discuss how the results would change if either of these assumptions were violated.

**Limitations:**

yes

**Strengths And Weaknesses:**

Strengths: The theoretical results appear to be technically sound and well supported. The overall presentation is clear, and the ideas are communicated effectively. In general, I find the extension to the setting of non-oblivious adversaries and contextual bandits both interesting and relevant.

Weaknesses: In my view, the main issue is that the paper does not clearly explain in what sense the results are novel and where the main technical contribution lies. Based on the presentation in the main paper, one might get the impression that the first part (extending Seldin et al. (2014) to contextual bandits with a non-oblivious adversary) is relatively straightforward, obtained by replacing the EXP3-based ideas with an EXP4-type algorithm and adapting the proofs accordingly. The resulting bounds also appear to be almost identical. A similar concern applies to the second part, where it seems that the analysis largely follows from combining the results for the two different $\eta$-regimes with the free-observation budget.

---

> ### Author Rebuttal · Authors · 2026-03-30
>
> Thanks for your suggestions.
>
> **Regarding the novelty and technical contribution:**
> We appreciate the reviewer's careful reading. Our work contributes three distinct technical novelties beyond adapting EXP3 to EXP4.
>
> *First, the lower bound construction.*
> We establish a minimax lower bound that shows the fundamental shift from $\Theta(T^{1/2})$ to $\Theta(T^{2/3})$ in the presence of costs. The proof in Lemma 2.2 requires a novel reduction from contextual bandits to multiple independent MAB instances, carefully accounting for the cost structure. This lower bound is new and is a prerequisite for understanding the problem's fundamental difficulty.
>
> *Second, the free-observation-budget analysis.* This setting is entirely new and does not appear in previous paid-observation literature. We discover a critical budget threshold and design a meta-controller (Algorithm 2) that switches between two specialized strategies. The design is non-trivial: in the large-budget regime, we propose a probabilistic budget allocation scheme (Algorithm 4, Appendix F) to achieve the required uniform observation probability; in the small-budget regime, we use Azuma's inequality to tightly control the size of the observation sets (see Lemma E.1, Page 23). The design and proofs are new.
>
> *Third, the oracle-efficient generalization (Section 4)*. This setting highlights the key difficulty of contextual bandits with paid observations under function approximation. Tucker et al. (2022) study linear contextual bandits but provide no theoretical guarantees. In contrast, our algorithm handles general function spaces, is computationally efficient via an online regression oracle, and provides rigorous regret bounds. Technical innovations include integrating the decision-estimation coefficient with cost-sensitive observations, convex-concave reformulations, application of Sion's minimax theorem, and closed-form observation probabilities, preserving optimal rates under cost constraints. See the main text on Page 7, lines 361-370.
>
> To clarify Algorithm 1's role: it establishes the baseline framework and the minimax lower bound, which are necessary foundations for the subsequent contributions. The free-budget analysis and oracle-efficient extension build on this foundation with substantially new algorithmic designs and proof techniques.
>
> To highlight our contributions, we revise our abstract as
>
> "
> We study contextual bandits with paid observations, where the learner actively chooses which actions to observe at a given cost in each round, with the goal of minimizing total regret that jointly accounts for learning loss and observation expenditure. We develop a near-optimal algorithm for adversarial environments and show that even small observation costs fundamentally raise the minimax regret order. We further uncover a novel phase transition under a free observation budget: below a critical threshold, free observations only reduce total cost without improving the regret rate; above it, asymptotic improvements become possible. To exploit this phenomenon, we design a meta-controller that adaptively switches between strategies to achieve near-optimal performance across all budget regimes. To handle large or infinite policy spaces, we also propose an oracle-efficient algorithm under a function approximation framework that maintains rigorous guarantees with computational efficiency. Our analysis also connects to related problems including switching costs, budgeted constraints, model misspecification, and knapsack bandits. Numerical experiments validate our theoretical findings.
> "
>
> We also follow your suggestions to clarify contributions in the main text.
> However, due to word limit, we cannot present all revisions here.
>
>
> **Regarding the constant-cost assumption in the free-budget setting:**
> The constant cost assumption is adopted *for simplicity* to clearly reveal the phase transition and its dependence on key parameters, particularly $c$. In this stylized setting, we are able to derive a clean threshold $B_0 = \Theta\left(c^{-2/3}\left(|\mathcal{A}|\ln|\Pi|\right)^{1/3}T^{2/3}\right)$ and show how the regret order changes across it.
>
> When costs vary over time and actions, we conjecture that the critical budget threshold generalizes to $B_0 = \Theta\left(\left(\sum_{t=1}^T \sqrt{\sum_{a} c_t(a)}\right)^{2/3}\right)$. The intuition is that $\sum_{t=1}^T \sqrt{\sum_{a} c_t(a)}$ already appears as the cost-dependent term in our general regret bound (Theorem 2.1), so the phase transition threshold should inherit this structure. If one additionally assumes that costs are i.i.d., one could estimate and predict future costs to optimize budget allocation, but this would require a substantially more complex design (e.g., online learning with cost predictions). We view this as a promising direction for future work.

---

> > ### Author Rebuttal · Reviewer_gBcu · 2026-04-02
> >
> > The rebuttal has adequately addressed my questions. Although I still view the theoretical contribution as somewhat modest, I generally have a positive assessment of the paper overall. I will keep my score unchanged.

---

> > > ### Author Response · Authors · 2026-04-04
> > >
> > > Thank you for your acknowledgment and for the positive overall assessment. We fully appreciate your constructive feedback throughout the process.

---

### Decision · Program_Chairs · 2026-04-30

**Decision:**

Accept (regular)

**Comment:**

The paper studied contextual bandits with paid observations, where the learner can actively acquire additional feedback at a cost. The paper appears to analyze a major challenge in such settings: balancing the benefit of acquiring additional information with the cost incurred by observing it. The adversarial environments are considered here. Author(s) are suggested to adopt reviews to further improve the manuscript.